# Revisiting Robustness for LLM Safety Alignment
# via Selective Geometry Control

**Yonghui Yang**[1][*]  **Wenjian Tao**[2][*][†]  **Jilong Liu**[2][*][†]  **Xingyu Zhu**[1]  **Junfeng Fang**[1]  **Weibiao Huang**[3]  **Le Wu**[2]
**Richang Hong**[2]  **Tat-Seng Chua**[1]

## Abstract

Safety alignment of large language models remains brittle under domain shift and noisy preference supervision. Most existing robust alignment methods focus on uncertainty in alignment data, while overlooking optimization-induced fragility in preference-based objectives. In this work, we revisit robustness for LLM safety alignment from an optimization geometry perspective, and argue that robustness failures cannot be addressed by data-centric methods alone. We propose *ShaPO*, a geometry-aware preference optimization framework that enforces worst-case alignment objectives via selective geometry control over alignment-critical parameter subspace. By avoiding uniform geometry constraints, ShaPO mitigates the over-regularization that can harm robustness under distribution shift. We instantiate ShaPO at two levels: token-level ShaPO stabilizes likelihood-based surrogate optimization, while reward-level ShaPO enforces reward-consistent optimization under noisy supervision. Across diverse safety benchmarks and noisy preference settings, ShaPO consistently improves safety robustness over popular preference optimization methods. Moreover, ShaPO composes cleanly with data-robust objectives, yielding additional gains and empirically supporting the proposed optimization-geometry perspective. The code is available at `https://github.com/liujilong0116/ShaPO`.

## 1. Introduction

Safety alignment of large language models (LLMs) is commonly approached through preference-based optimization, where models are trained to prefer safe responses over unsafe ones based on human judgments (Ji et al., 2023c; Lu et al., 2025). This paradigm is most notably instantiated through Reinforcement Learning from Human Feedback (RLHF) (Christiano et al., 2017; Ouyang et al., 2022), which has become a standard approach for aligning LLM behavior with human safety preferences. To reduce the complexity and instability of RLHF, recent methods such as Direct Preference Optimization (DPO) (Rafailov et al., 2023) and its variants (Gheshlaghi Azar et al., 2023; Chowdhury et al., 2024; Mitchell, 2023; Meng et al., 2024) directly optimize preference objectives, substantially simplifying training while retaining competitive alignment performance. As a result, preference-based optimization has emerged as a practical and widely adopted paradigm for safety alignment.

Despite their effectiveness during training, preference-aligned models often exhibit brittle behavior in deployment, suggesting that safety alignment can remain shallow and insufficiently robust (Anwar et al., 2024; Qi et al., 2024). Empirical studies show that models performing well on curated safety benchmarks may still produce unsafe responses under domain shift or noisy preference supervision (Wei et al., 2024; Gao et al., 2024), exposing a gap between training- and deployment-time robustness. To address this gap, prior work has primarily focused on reducing data uncertainty, through strategies such as preference data selection (Gao et al., 2025; Kong et al., 2024; Yeh & Li, 2025), margin-based reweighting (Gheshlaghi Azar et al., 2023; Huang et al., 2025), and distributionally robust optimization over preference distributions (Wu et al., 2024; Zhu et al., 2025a). While effective against annotation noise and distribution shift, these data-centric approaches implicitly attribute robustness failures to uncertainty in the supervision signal.

From the robustness perspective, existing data-centric approaches primarily target uncertainty in the supervision signal, while the optimization geometry determines how sensitively alignment behavior responds to such uncertainty

---

[*]Equal contribution  [†]Work done during internship at NExT++ Research Centre, National University of Singapore. [1]National University of Singapore [2]Hefei University of Technology [3]ST Engineering Ltd., Singapore. Correspondence to: Richang Hong <hongrh.hfut@gmail.com>.

*Proceedings of the 43rd International Conference on Machine Learning*, Seoul, South Korea. PMLR 306, 2026. Copyright 2026 by the author(s).

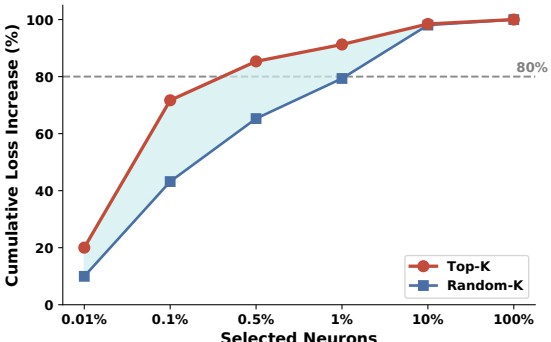

*Figure 1.* Cumulative contribution to worst-case alignment loss under parameter perturbations. We compare the fraction of the total worst-case loss increase accounted for by perturbing probe-identified safety-critical neurons (Top-K) versus randomly selected neurons of the same size (Random-K).

during training. In particular, sharp alignment landscapes can amplify even minor noise in preference data, leading to fragile safety behavior under domain shift (Foret et al., 2020; Wang et al., 2023a). To this end, we revisit robustness in safety alignment from the lens of optimization geometry. Nevertheless, introducing geometry control to preference-based alignment still presents practical questions about *where* constraints should be applied, *at what level* they should be instantiated, and *how* they interact with data-centric robustness methods.

**Selective geometry control.** We propose *Sharpness-aware Preference Optimization (ShaPO)* , a geometry-aware framework that enforces worst-case alignment stability only along alignment-critical directions. Prior work suggests that safety behavior in large language models is localized to a small subset of internal representations (Wei et al., 2024; Zhao et al., 2025b), making uniform geometry control prone to over-constraining safety-irrelevant parameters and degrading cross-domain robustness (Niu et al., 2025; Perin et al., 2025). To identify where worst-case alignment sensitivity arises, we conduct an empirical study ranking neurons by similarity to probe-identified safety-critical directions. As shown in Figure 1, perturbing fewer than 0.5% of these neurons accounts for over 80% of the cumulative worst-case alignment loss, while random neurons of the same size contribute substantially less. This concentration indicates that selective geometry control can approximate worst-case optimization without unnecessarily restricting non-safety parameters, preserving degrees of freedom important for robust generalization under domain shift.

**Multi-level instantiations.** Preference-based alignment objectives exhibit distinct failure modes at different abstraction levels, motivating geometry-aware robustness at both token and reward levels. *Token-level ShaPO* applies selective

geometry control to likelihood-based surrogate alignment objectives, such as the DPO loss (Rafailov et al., 2023), without an explicit reward model. This instantiation provides a lightweight mechanism for stabilizing preference optimization driven by token-level supervision. *Reward-level ShaPO* applies selective geometry control to reward-based alignment objectives, using a pre-trained reward model as a semantically grounded safety proxy. By enforcing stability with respect to reward-consistent signals, this instantiation offers an alternative robustness pathway where token-level surrogates may be misaligned with semantic safety.

**Composability.** ShaPO targets optimization-induced fragility rather than supervision uncertainty, and is therefore complementary to data-centric robustness techniques. This scope allows geometry-aware robustness to be combined with existing data-robust objectives without modifying the supervision signal. In experiments across diverse safety benchmarks under domain shift and noisy supervision, both token-level and reward-level variants of ShaPO can be applied either alone or alongside representative data-centric methods, yielding additional robustness gains when combined. These results support optimization geometry as a complementary design dimension for robust LLM safety alignment.

**Conflict of Interest Disclosure.** The authors declare no financial conflicts of interest.

## 2. Preliminaries

**Safety Alignment via Preference Optimization.** Safety alignment seeks to ensure that LLMs avoid generating harmful outputs while maintaining task utility. In practice, this is typically achieved through post-training preference optimization, where models are trained to favor safer responses over unsafe alternatives (Ouyang et al., 2022; Ji et al., 2023a). This formulation underlies a broad class of alignment methods, including reinforcement learning from human feedback (RLHF (Bai et al., 2022)) and its simplified variants, such as Direct Preference Optimization (DPO (Rafailov et al., 2023)). Concretely, given a dataset of preference triples $\mathcal{O} = \{(x, y^w, y^l)\}$, where $y^w$ is preferred over $y^l$, DPO instantiates preference-based alignment by optimizing the following objective:

$$\mathcal{L}_{\text{align}}(\theta) = -\mathbb{E}_{(x,y^w,y^l)\sim\mathcal{O}}\big[\log\sigma\big(r_\theta(x,y^w) - r_\theta(x,y^l)\big)\big],$$
(1)

where $r_\theta(x,y) = \beta\log\frac{\pi_\theta(y|x)}{\pi_{\text{ref}}(y|x)}$ denotes an implicit reward function. More generally, we denote the preference-based alignment objective as $\mathcal{L}_{\text{align}}(\theta)$.

**Data-uncertainty Robustness.** While the preference-based alignment pipeline provides a principled objective

for safety alignment, its effectiveness depends critically on the quality and reliability of the underlying preference supervision. In practice, preference data may be noisy, inconsistent, or subject to distribution shift, leading to misspecified optimization targets (Wei et al., 2024; Gao et al., 2024). A common approach to address such uncertainty is to adopt a distributionally robust optimization (Delage & Ye, 2010; Levy et al., 2020) formulation, which optimizes alignment performance under worst-case perturbations of the preference data. Formally, the optimization objective can be written as

$$\min_{\theta} \max_{\mathcal{O}'} \mathcal{L}_{\text{align}}(\mathcal{O}'; \theta), \quad s.t. \mathbb{D}_{\phi}(\mathcal{O}', \mathcal{O}) \leq \eta, \quad (2)$$

where $\mathcal{O}$ denotes the observed preference supervision, and $\mathcal{O}'$ represents a perturbed preference distribution within a neighborhood defined by the divergence $\mathbb{D}_{\phi}(\mathcal{O}', \mathcal{O}) \leq \eta$. This formulation has been widely adopted to model robustness with respect to worst-case variations in preference supervision, and underlies a class of data-centric robust alignment approaches that address noise and distribution shift in preference data (Wu et al., 2024; Zhu et al., 2025a).

**Geometry-related Robustness.** Data-centric robustness formulations improve alignment reliability by accounting for uncertainty in preference supervision. However, such approaches implicitly attribute robustness failures solely to variations in the data, overlooking the sensitivity that may arise from the optimization process itself (Jiang et al., 2019; Foret et al., 2020). To capture robustness beyond data uncertainty, we consider a formulation that accounts for both data-side and parameter-side sources of fragility in preference-based alignment. For conceptual clarity, robustness in preference-based alignment can be viewed along two complementary axes:

$$\min_{\theta} \underbrace{\max_{\mathcal{O}':\mathbb{D}_{\phi}(\mathcal{O}',\mathcal{O})\leq\eta} \mathcal{L}_{\text{align}}(\theta; \mathcal{O}')}_{\text{data-side}} + \underbrace{\max_{\|\epsilon\|\leq\rho} \mathcal{L}_{\text{align}}(\theta + \epsilon; \mathcal{O})}_{\text{parameter-side}},$$

$$(3)$$

where $\epsilon$ denotes a bounded perturbation in the model parameter space. Given the above analysis, we focus on parameter-side robustness to improve the stability of safety alignment, viewing *optimization geometry* as a complementary axis to existing data-uncertainty approaches.

## 3. Selective Geometry Control for Robust Safety Alignment

We present ShaPO , a geometry-aware optimization framework for robust safety alignment. Figure 2 provides an overview of the framework, and this section details the selective geometry control objective, safety-critical subspace approximation, and its instantiations at both the token and reward levels.

### 3.1. Optimization Objective of ShaPO

Preference-based alignment specifies relative behavioral constraints, and robustness requires these preferences to remain stable under small parameter perturbations. We adopt a geometry-aware formulation that optimizes the alignment objective under worst-case perturbations:

$$\max_{\|\epsilon\|\leq\rho} \mathcal{L}_{\text{align}}(\theta + \epsilon), \quad (4)$$

where $\rho$ denotes the perturbation radius. Rather than enforcing robustness uniformly across all parameters, ShaPO restricts adversarial perturbations to a safety-critical parameter subspace. Let $\mathcal{S} \subset \mathbb{R}^d$ denote this subspace, selective geometry control is defined as

$$\mathcal{L}_{\text{ShaPO}}(\theta) = \max_{\|\epsilon_{\mathcal{S}}\|\leq\rho} \mathcal{L}_{\text{align}}(\theta + \epsilon_{\mathcal{S}}), \quad (5)$$

where $\epsilon_{\mathcal{S}}$ denotes perturbations constrained to $\mathcal{S}$. This objective enforces robustness of the alignment loss with respect to safety-critical directions.

### 3.2. Safety-critical Subspace Approximation

The safety-critical subspace $\mathcal{S}$ in Eq. (5) is approximated using a lightweight probing-based signal that identifies directions most relevant to safety behavior. Given a representation $h \in \mathbb{R}^d$ from the residual stream of the final transformer layer, we train a linear probe $p \in \mathbb{R}^d$ by minimizing the logistic loss:

$$\min_{p} \mathbb{E}_{(h,y)}\Big[ \log\big(1 + \exp(-y\langle p, h\rangle)\big)\Big], \quad (6)$$

where $y \in \{+1, -1\}$ indicates whether the corresponding response is labeled as safe or unsafe. The resulting probe direction captures variations in representation space that are predictive of safety behavior.

To localize safety-critical parameters, we measure the alignment between $p$ and internal parameter-induced representations. Following prior mechanistic analyses (Lee et al., 2024), we focus on value vectors $\{v_j\}$ in transformer blocks and compute similarity scores:

$$s_j = \big|\langle v_j, p\rangle\big|. \quad (7)$$

Parameters $\theta_{\mathcal{S}} = \text{TopK}\{v_j\}$ with the highest similarity scores define the safety-critical subspace $\mathcal{S}$ used for selective perturbation. The same probe-based Top-$K$ selection procedure is used to produce the empirical analysis shown in Figure 1, ensuring consistency between the motivating evidence and the selective geometry control objective. We conservatively select Top-1% neurons to cover over 90% of worst-case loss, trading slight redundancy for stability. Detailed probe training procedure and decoded results are provided in Appendix E.1.

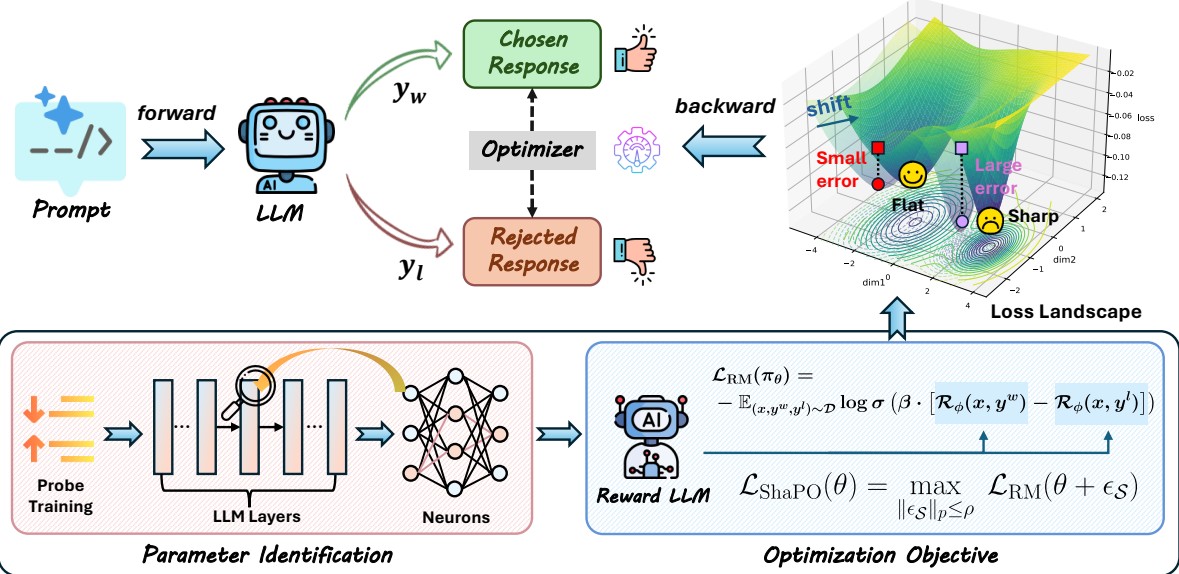

*Figure 2.* Overview of ShaPO , a robust preference optimization framework with selective geometry control. ShaPO minimizes the worst-case alignment loss under adversarial perturbations restricted to the alignment-critical parameter subspace. Here is the instantiation of ShaPO at reward level.

### 3.3. Instantiations of ShaPO

Given the estimated safety-critical subspace $\mathcal{S}$, we instantiate ShaPO by applying selective geometry control to concrete preference-based alignment objectives. The underlying principle remains the same: robustness is enforced by restricting worst-case parameter perturbations to $\mathcal{S}$, while the specific form of the alignment loss determines the level at which preferences are modeled. We consider two representative instantiations at the token and reward levels, respectively.

① **Token-level ShaPO .** We first instantiate ShaPO with the token-level objectives, such as DPO (Rafailov et al., 2023), rDPO (Chowdhury et al., 2024), Dr.DPO (Wu et al., 2024) and so on, which serves as a likelihood-based surrogate for preference alignment. Taking DPO loss as an example, token-level ShaPO applies selective geometry control by optimizing the worst-case DPO loss under perturbations restricted to parameters $\epsilon_{\mathcal{S}}$:

$$\mathcal{L}_{\text{ShaPO}}(\theta) = \max_{\|\epsilon_{\mathcal{S}}\|_p \leq \rho} \mathcal{L}_{\text{DPO}}(\theta + \epsilon_{\mathcal{S}})$$

$$= \max_{\|\epsilon_{\mathcal{S}}\|_p \leq \rho} -\mathbb{E} \log \sigma(\beta \cdot [\sum_{t=1}^{T_w} \log \frac{\pi_{\theta+\epsilon_{\mathcal{S}}}(y_t^w \mid x, y_{<t}^w)}{\pi_{\text{ref}}(y_t^w \mid x, y_{<t}^w)}$$

$$- \sum_{t=1}^{T_l} \log \frac{\pi_{\theta+\epsilon_{\mathcal{S}}}(y_t^l \mid x, y_{<t}^l)}{\pi_{\text{ref}}(y_t^l \mid x, y_{<t}^l)}]),$$

(8)

where $T_w$ and $T_l$ denote the length of the chosen response and the rejected response. This token-level ShaPO encour-

ages the model to remain preference-consistent under local parameter perturbations within the identified sensitive subspace.

② **Reward-level ShaPO .** While token-level ShaPO improves robustness, its objective remains tied to token likelihoods, which may not faithfully reflect semantic preferences (Yang et al., 2023; Li et al., 2025). To address this limitation, we introduce *reward-level ShaPO* , which applies selective geometry control to a reward-based alignment objective.

Let $R_\phi(y \mid x)$ be a pretrained reward model that outputs a scalar preference score for response $y$ given prompt $x$. For each preference pair $(y^w, y^l)$, we derive a soft preference target:

$$t(x, y^w, y^l) = \sigma\big(\beta_r[R_\phi(y^w \mid x) - R_\phi(y^l \mid x)]\big), \quad (9)$$

where $\beta_r$ controls the sharpness of the reward gap. The policy predicts a preference probability:

$$p_\theta(y^w \succ y^l \mid x) = \sigma\big(s_\theta(x, y^w) - s_\theta(x, y^l)\big), \quad (10)$$

with $s_\theta(x, y)$ denoting the model score (e.g., log-likelihood). The reward-based alignment loss is given by the binary cross-entropy:

$$\mathcal{L}_{\text{RM}}(\theta) = -\big[t \log p_\theta + (1 - t) \log(1 - p_\theta)\big]. \quad (11)$$

Reward-level ShaPO then enforces robustness by optimizing the worst-case reward-based loss under selective perturbations:

$$\mathcal{L}_{\text{ShaPO}}(\theta) = \max_{\|\epsilon_{\mathcal{S}}\|_p \leq \rho} \mathcal{L}_{\text{RM}}(\theta + \epsilon_{\mathcal{S}}). \quad (12)$$

**Discussion of reward-level ShaPO .** Reward-level ShaPO offers several complementary advantages for robust alignment. First, by anchoring optimization to semantic preference signals from a pretrained reward model, the alignment objective is defined at the preference-probability level rather than token likelihoods, making it more compatible with parameter perturbations such as SAM and encouraging semantic consistency under perturbation. Second, by simulating parameter perturbations within a reward-based objective, reward-level ShaPO establishes a structural connection between offline preference optimization and online RLHF, while remaining fully offline and avoiding costly on-policy rollouts. Third, from a data-centric perspective, the use of an external reward model can mitigate noise and ambiguity in token-level supervision, providing a complementary source of robustness under uncertain or noisy annotations.

### 3.4. Optimization Process

Given the instantiated alignment objective $\mathcal{L}_{\text{align}}(\theta)$, ShaPO adopts a sharpness-aware optimization scheme inspired by SAM (Foret et al., 2020) to approximate locally robust solutions. The training procedure consists of two forward–backward passes. In the first pass, we compute the gradient with respect to the selected parameter subspace $\theta_{\mathcal{S}}$:

$$g_{\mathcal{S}} = \nabla_{\theta_{\mathcal{S}}} \mathcal{L}_{\text{align}}(\theta). \tag{13}$$

Based on this gradient, we construct a normalized perturbation:

$$\epsilon_{\mathcal{S}} = \rho \cdot \frac{g_{\mathcal{S}}}{\|g_{\mathcal{S}}\|_2 + \varepsilon}, \tag{14}$$

where $\rho$ controls the perturbation radius and $\varepsilon$ is a small constant for numerical stability. This perturbation lies on the surface of an $L_2$ ball centered at $\theta_{\mathcal{S}}$.

In the second pass, the model parameters are updated by evaluating the alignment objective under the perturbed parameters:

$$\theta \leftarrow \theta - \eta \cdot \nabla_{\theta} \mathcal{L}_{\text{align}}(\theta + \epsilon_{\mathcal{S}}). \tag{15}$$

This two-step optimization approximates a local worst-case perturbation in the specified parameter subspace, encouraging convergence toward flatter minima with respect to $\theta_{\mathcal{S}}$. To reduce computational overhead, the sharpness-aware update is performed once every $T$ training steps rather than at every iteration. The complete training algorithm is provided in Appendix A.

**Robustness Insights of ShaPO .** ShaPO optimizes the alignment loss under worst-case parameter perturbations restricted to the safety-critical subspace $\mathcal{S}$. For a differentiable alignment loss $\mathcal{L}(\theta)$ and sufficiently small perturbation radius $\rho$, a second-order expansion suggests:

$$\max_{\|\epsilon_{\mathcal{S}}\| \leq \rho} \mathcal{L}(\theta + \epsilon_{\mathcal{S}}) \approx \mathcal{L}(\theta) + \rho \|\nabla_{\mathcal{S}} \mathcal{L}(\theta)\|_2 + \frac{\rho^2}{2} \lambda_{\max}(H_{\mathcal{S}}),$$

where $H_{\mathcal{S}}$ denotes the Hessian restricted to $\mathcal{S}$. This indicates that ShaPO implicitly suppresses both large alignment gradients and high curvature along safety-critical directions. Since noisy or conflicting preference supervision often induces unstable updates in these directions, selectively controlling the worst case improves the robustness of safety behavior without overly constraining capability-related parameters. Based on the previous study (Lee et al., 2024), we define the *bypass boundary* as the radius within which parameter perturbations do not lead to unsafe behavior. For a risk threshold $\tau$, the boundary $\delta$ satisfies:

$$\delta \lesssim \sqrt{\frac{2(\tau - \mathcal{L}(\theta))}{\lambda_{\max}(H_{\mathcal{S}})}}.$$

As ShaPO reduces the subspace curvature $\lambda_{\max}(H_{\mathcal{S}})$, it effectively expands this boundary, implying stronger safety guarantees under parameter perturbations. Detailed analysis is provided in Appendix C.

## 4. Experiments

In this section, we evaluate whether geometry-aware optimization improves the robustness of LLM safety alignment. We assess ShaPO under three settings: in-distribution alignment, cross-domain generalization, and robustness to noisy preference supervision. We compare ShaPO with standard preference optimization and data-centric robust methods, and conduct targeted ablations to analyze the role of selective geometry control.

### 4.1. Experimental Setting

**Benchmarks and Backbones.** We train alignment models on the PKU-SafeRLHF-30K (Ji et al., 2023a) dataset and evaluate the robustness of ShaPO across five safety benchmarks: PKU-SafeRLHF-30K, HH-RLHF-Safety (Bai et al., 2022), Do-Not-Answer (Wang et al., 2023b), Harm-Bench (Mazeika et al., 2024), and SaladBench (Li et al., 2024). These benchmarks cover diverse safety scenarios and enable evaluation of both in-distribution performance and cross-domain generalization. To assess robustness under noisy preference supervision, we follow the protocol of (Wu et al., 2024) and introduce controlled label flips on the training data. Models are trained on corrupted preference data while evaluated on clean validation sets, allowing us to isolate the impact of noisy supervision on alignment robustness. We conduct experiments on a diverse set of backbone models spanning different families and scales, including Pythia-2.8B [1], LLaMA-3.2-3B [2], LLaMA-3-8B [3],

---

[1] https://huggingface.co/EleutherAI/pythia-2.8b
[2] https://huggingface.co/meta-llama/Llama-3.2-3B
[3] https://huggingface.co/meta-llama/Meta-Llama-3-8B

*Table 1.* Safety alignment performances on the IID setting. All methods are trained and evaluated on the PKU-SafeRLHF-30K dataset, and we report WinRate and two safety judge scores.

| Methods | Pythia-2.8B | | | LLaMA-3.2-3B | | | LLaMA-3-8B | | | Qwen2.5-7B | | |
|---|---|---|---|---|---|---|---|---|---|---|---|---|
| | WR↑ | MD↓ | NV↓ | WR↑ | MD↓ | NV↓ | WR↑ | MD↓ | NV↓ | WR↑ | MD↓ | NV↓ |
| Vanilla | 40.18% | 73.31% | 26.94% | 44.93% | 66.79% | 18.42% | 42.82% | 67.54% | 24.69% | 69.09% | 32.21% | 13.16% |
| SFT | 41.05% | 71.30% | 26.32% | 35.93% | 68.67% | 45.36% | 23.85% | 69.92% | 52.51% | 25.09% | 70.55% | 52.88% |
| DPO | 44.23% | 57.14% | 19.17% | 73.70% | 11.28% | 6.14% | 69.19% | 26.57% | 14.41% | 75.81% | 20.55% | 7.27% |
| IPO | 45.27% | 61.65% | 22.68% | 72.70% | 11.78% | 6.14% | 70.22% | 26.82% | 13.53% | 73.97% | 21.30% | 8.02% |
| cDPO | 45.27% | 60.28% | 16.92% | 72.91% | 25.44% | 12.78% | 59.48% | 37.94% | 23.43% | 67.65% | 31.83% | 13.66% |
| rDPO | 46.54% | 57.77% | 23.43% | 81.30% | 4.14% | 1.75% | 82.13% | 6.64% | 3.26% | 82.74% | 4.51% | 2.01% |
| Dr.DPO | 52.66% | 29.95% | 18.17% | 83.77% | 2.38% | 0.50% | 82.50% | 4.39% | 0.50% | 88.93% | 2.01% | 0.13% |
| *ShaPO-T* | 56.14% | 15.16% | **1.76%** | **85.88%** | **1.25%** | **0.13%** | 85.45% | **2.76%** | **0.13%** | **89.26%** | **0.75%** | 0.13% |
| *ShaPO-R* | **57.14%** | **11.40%** | 2.76% | 85.29% | 1.75% | 0.50% | **87.45%** | 1.13% | **0.13%** | 89.23% | 1.13% | **0.00%** |

and Qwen2.5-7B [4]. For the implementation of *reward-level ShaPO*, we use a single safety-oriented preference model, *PKU-Alignment/beaver-7b-v1.0-cost* [5], trained on the PKU-SafeRLHF dataset.

**Baselines and Evaluation Metrics.** We conduct experiments on diverse LLM backbones and compare ShaPO with a set of representative preference-based alignment methods across all evaluation settings, including cross-domain generalization and robustness to noisy preference supervision. Specifically, we include the Vanilla LLM, supervised fine-tuning (SFT) (Ouyang et al., 2022), and Direct Preference Optimization (DPO) (Rafailov et al., 2023) as standard alignment baselines. In parallel, we further compare with several robust preference optimization variants designed to mitigate noise and distributional uncertainty, including IPO (Gheshlaghi Azar et al., 2023), cDPO (Mitchell, 2023), rDPO (Chowdhury et al., 2024), and Dr.DPO (Wu et al., 2024).

All baseline methods are evaluated on the following widely used safety metrics: *Win Rate* (WR, the higher the safer); *Attack Success Rate* (ASR, the lower the safer). Specifically, ASR is evaluated by two widely used safety judges: MD-Judge-v0_2-internlm2_7b (MD) [6] and NVIDIA Llama-3.1-Nemotron-Safety-Guard-8B-v3 [7]. Detailed experimental settings are introduced on Appendix D.

### 4.2. In-distribution Alignment Performance

We first evaluate the alignment performance of ShaPO under the in-distribution (IID) setting on PKU-SafeRLHF-30K, with results summarized in Table 1. While our primary focus is robustness under distribution shift and noisy supervi-

vision, this experiment serves as a sanity check to verify that geometry-aware optimization does not compromise standard alignment quality. Across all backbones, *ShaPO-T* consistently achieves the highest or near-highest WR, while substantially reducing safety risks measured by both MD and NV judges. Compared to DPO and its data-robust variants, *ShaPO-T* improves WR by a clear margin on all four models, while simultaneously lowering Attack Success Rates, indicating that geometry-aware regularization does not induce overly conservative behavior. Notably, these gains are consistent across model families and scales, suggesting that the benefits of selective geometry control are not architecture-specific. *ShaPO-R* further improves safety metrics on larger backbones, achieving the lowest MD and NV scores on LLaMA-3-8B and Qwen2.5-7B. Although reward-level ShaPO does not optimize token-level likelihood directly, it achieves safety violation rates comparable to token-level ShaPO under clean IID supervision, while consistently outperforming DPO and Dr.DPO. This suggests that enforcing geometry control with respect to semantically grounded reward signals can be an effective alternative to token-level surrogates, without sacrificing in-domain alignment performance. Overall, these results demonstrate that geometry-aware optimization preserves, and often improves, in-distribution alignment performance while significantly enhancing safety. Importantly, the absence of a trade-off between WR and safety metrics indicates that the robustness gains observed in later experiments do not stem from degraded alignment quality under standard settings.

### 4.3. Safety robustness under domain shift

We next evaluate the robustness of ShaPO under out-of-distribution (OOD) safety benchmarks, with results reported in Table 2. All models are trained on PKU-SafeRLHF-30K and evaluated on four heterogeneous safety benchmarks, including Do-Not-Answer, HarmBench, HH-RLHF, and SaladBench, which exhibit substantial distributional differences from the training data. Across both LLaMA-3-8B and Qwen2.5-7B backbones, *ShaPO* consistently achieves the

---

[4]https://huggingface.co/Qwen/Qwen2.5-7B

[5]https://huggingface.co/PKU-Alignment/beaver-7b-v1.0-cost

[6]https://huggingface.co/OpenSafetyLab/MD-Judge-v0_2-internlm2_7b

[7]https://huggingface.co/nvidia/Llama-3.1-Nemotron-Safety-Guard-8B-v3

*Table 2.* Safety alignment performances on the domain shift setting, which were evaluated on four safety benchmarks with two judges. The reported results are conducted on LLaMA-3-8B and Qwen2.5-7B backbones.

| Backbones | Methods | Do-Not-Answer | | HarmBench | | HH-RLHF | | SaladBench | | AVG. |
|---|---|---|---|---|---|---|---|---|---|---|
| | | MD↓ | NV↓ | MD↓ | NV↓ | MD↓ | NV↓ | MD↓ | NV↓ | |
| LLaMA-3-8B | Vanilla | 54.58% | 12.37% | 87.50% | 23.50% | 65.10% | 19.92% | 73.14% | 23.26% | 46.27% |
| | SFT | 53.52% | 33.48% | 93.50% | 81.50% | 69.27% | 47.54% | 75.10% | 58.68% | 64.03% |
| | DPO | 14.18% | 7.78% | 41.00% | 28.50% | 23.05% | 10.30% | 31.55% | 19.75% | 22.85% |
| | IPO | 14.29% | 7.36% | 33.50% | 17.50% | 21.70% | 8.30% | 30.12% | 15.69% | 20.42% |
| | cDPO | 22.39% | 13.86% | 61.00% | 44.00% | 30.32% | 13.34% | 42.53% | 26.44% | 30.7% |
| | rDPO | 1.17% | 0.53% | 6.00% | 1.5% | 6.88% | 2.25% | 5.38% | 2.38% | 3.97% |
| | Dr.DPO | 0.96% | **0.00%** | 2.00% | **0.00%** | 4.58% | 0.37% | 3.52% | 0.51% | 2.09% |
| | *ShaPO-T* | **0.43%** | 0.11% | **0.50%** | **0.00%** | 2.94% | 0.19% | 2.48% | 0.23% | 1.37% |
| | *ShaPO-R* | 0.53% | **0.00%** | **0.50%** | **0.00%** | **1.86%** | 0.12% | **1.13%** | 0.08% | **0.70%** |
| Qwen2.5-7B | Vanilla | 21.11% | 7.78% | 44.50% | 27.00% | 32.52% | 10.76% | 33.85% | 13.77% | 23.02% |
| | SFT | 52.67% | 31.56% | 95.00% | 74.00% | 70.54% | 48.01% | 73.45% | 55.05% | 62.37% |
| | DPO | 10.98% | 4.16% | 32.50% | 14.00% | 17.70% | 4.90% | 22.97% | 8.63% | 14.39% |
| | IPO | 10.34% | 3.41% | 30.50% | 14.00% | 18.02% | 5.03% | 22.16% | 8.24% | 14.00% |
| | cDPO | 16.42% | 5.76% | 45.50% | 19.50% | 26.13% | 8.59% | 32.56% | 13.66% | 21.26% |
| | rDPO | 1.17% | 0.32% | 3.00% | 1.00% | 6.14% | 1.76% | 4.23% | 1.76% | 3.18% |
| | Dr.DPO | 0.21% | **0.00%** | 0.50% | **0.00%** | 2.39% | 0.22% | 1.53% | 0.46% | 1.05% |
| | *ShaPO-T* | 0.11% | **0.00%** | **0.00%** | **0.00%** | **1.09%** | 0.12% | **0.68%** | 0.07% | **0.42%** |
| | *ShaPO-R* | **0.00%** | **0.00%** | 0.50% | **0.00%** | 1.31% | 0.12% | 0.98% | 0.06% | 0.55% |

lowest ASR across nearly all benchmarks and safety judges. Compared with standard preference optimization methods (e.g., DPO) and data-centric robust variants (IPO, cDPO, rDPO, and Dr.DPO), *ShaPO-T* substantially reduces ASR, indicating improved robustness to domain shift. Notably, these gains are consistent across diverse safety scenarios, suggesting that geometry-aware optimization generalizes beyond the specific characteristics of any single benchmark.

Reward-level *ShaPO-R* further strengthens safety robustness, achieving the lowest average ASR on both backbones. In particular, *ShaPO-R* consistently outperforms Dr.DPO and rDPO on challenging benchmarks such as HH-RLHF and SaladBench, where preference distributions and safety criteria differ most from the training data. This highlights the benefit of directly aligning optimization geometry with semantic preference signals at the reward level, resulting in stronger robustness under distribution shift. Overall, the OOD results demonstrate that robustness failures in safety alignment cannot be fully addressed by data-centric techniques alone. By explicitly controlling optimization geometry, ShaPO provides complementary and often larger robustness gains, leading to more stable safety behavior when deployed beyond the training distribution. To further examine robustness under more challenging settings, we report additional results on smaller backbones in Appendix E.3.

### 4.4. Safety robustness to noisy supervision

We further evaluate the robustness of ShaPO under noisy preference supervision by introducing controlled label flips during training, while evaluating all models on clean validation data. Figure 5 illustrates the Win Rate metric of all methods across increasing flip rates, ranging from 0% to 40%. As the noise level increases, standard preference optimization methods such as DPO exhibit a pronounced degradation, indicating high sensitivity to corrupted supervision. Data-centric robust variants mitigate this degradation to some extent; however, their performance still deteriorates notably under severe noise. In contrast, *ShaPO-T* consistently maintains a higher win rate across all noise levels, demonstrating improved stability under noisy supervision.

Notably, *ShaPO-R* achieves the strongest robustness, yielding the highest performance under all flip rates. Under severe noise (40% flipped preferences), *ShaPO-R* substantially outperforms both standard and data-robust baselines. This advantage stems from its use of an external reward model that provides a stable and noise-resilient supervision signal, which is decoupled from the corrupted preference labels used during training. By anchoring optimization to this independent reward signal, *ShaPO-R* effectively mitigates the influence of noisy human preferences and prevents error accumulation under high flip rates.

### 4.5. Comprehensive Analysis of ShaPO

**Composability of ShaPO.** Beyond serving as a standalone robustness mechanism, ShaPO is designed as a geometry-aware optimization framework that is complementary to existing data-centric alignment methods. To evaluate its composability, we integrate ShaPO with other

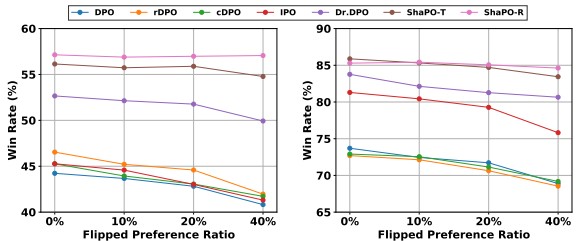

Figure 3. Sensitivities to noise environments, we compare the win rate of all methods across different preference data flipping ratios (10%, 20%, 40%). The left is the results on Pythia-2.8B, and the right is on LLaMA-3.2-3B backbone.

Table 3. ASR comparison between uniform, random, and selective geometry control on LLaMA-3-8B. IID results are evaluated on PKU-SafeRLHF-30K, while OOD results report the average ASR across four OOD safety benchmarks.

| Geometry Control | IID | | OOD (Avg.) | |
|---|---|---|---|---|
| | MD↓ | NV↓ | MD↓ | NV↓ |
| No Control | 26.57% | 14.41% | 28.78% | 16.91% |
| Random Control | 26.89% | 14.34% | 28.56% | 16.87% |
| Uniform Control | **25.56%** | **13.03%** | 26.83% | 15.12% |
| Selective Control | 27.32% | 13.91% | **26.31%** | **14.12%** |

data-centric alignment objectives, including DPO and its variants, and compare the resulting alignment performance. As shown in Figure 4, incorporating ShaPO consistently improves alignment robustness across different base objectives. Notably, these gains are observed on top of methods that already employ data-centric robustness strategies, indicating that geometry-aware robustness targets a distinct source of alignment fragility. Rather than replacing existing approaches, ShaPO provides additional stability by reducing sensitivity to parameter perturbations during optimization. The consistent improvements across diverse objectives suggest that robustness failures in safety alignment cannot be fully addressed by data-level interventions alone. By operating along an orthogonal optimization geometry axis, ShaPO composes naturally with data-centric techniques, yielding additive robustness gains.

**Why selective Geometry Control.** Table 3 systematically compares geometry control strategies — no control (degenerating to standard DPO), random, uniform, and selective — on LLaMA-3-8B with DPO loss, reporting ASR under both IID and OOD settings. This comparison isolates not whether geometry regularization helps, but where and how it should be applied. Under IID evaluation, all strategies yield comparable ASR, indicating that excessive optimization sensitivity does not manifest when training and evaluation distributions are well matched. Clear differences emerge under OOD evaluation: uniform control consistently yields higher ASR than selective control, while random control provides only marginal improvement, suggesting that unprincipled

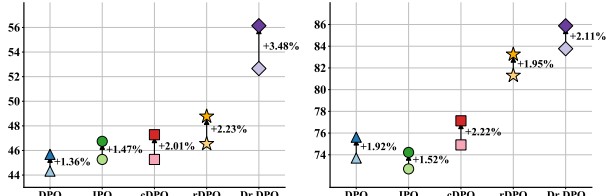

Figure 4. Composability of ShaPO with DPO and other data-centric alignment methods. We report the Win Rate compared with the chosen response; the left is comparisons on Pythia-2.8B, and the right is on the LLaMA-3.2-3B backbone.

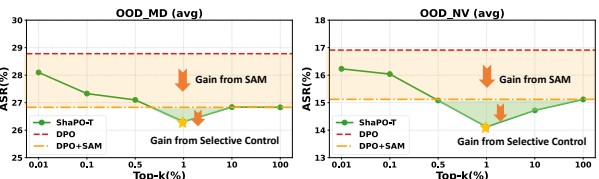

Figure 5. Effect of Selective Neuron Ratio. We compare ShaPO with different selective neuron ratios on LLaMA-3-8B, and report average ASR across four OOD safety benchmarks.

regularization fails to address the underlying sources of robustness failures. By contrast, selective geometry control achieves the lowest OOD ASR, demonstrating more reliable generalization across heterogeneous safety benchmarks.

Figure 5 illustrates how the selective neuron ratio (Top-k%) affects alignment robustness. As the ratio increases from 0.01% to 1%, ASR consistently decreases and reaches its minimum at Top-1%, indicating that a moderately sized neuron subset is sufficient to capture the most safety-critical geometric structures. Beyond this point, ASR gradually rises, suggesting that incorporating excessive neurons introduces redundant geometric information that undermines robustness. This non-monotonic trend reveals that robustness gains stem not from geometry regularization alone, but from applying it to alignment-critical directions. Uniform control suppresses both safety-critical and domain-adaptive components essential for handling distributional variation, whereas selective geometry control strikes a better balance between sharpness reduction and alignment preservation.

**Geometry of the Loss Landscape.** Table 4 reports the maximum Hessian eigenvalue (Max HE) and win rate across methods on LLaMA-3-8B. A consistent trend emerges: as methods progress from Base to DPO, Dr.DPO, and ShaPO-T, Max HE decreases monotonically from 25.2336 to 6.2466, indicating progressively flatter loss landscapes along the safety subspace. Correspondingly, win rate improves from 42.82% to 85.45%, confirming that lower curvature is associated with stronger alignment performance. These results suggest that safety alignment benefits from reducing parameter sensitivity along alignment-relevant directions, and that ShaPO-T achieves superior robustness by selectively

*Table 4.* Maximum Hessian Eigenvalue (Max HE) and Win Rate across methods on LLaMA-3-8B.

| Metric | Base | DPO | Dr.DPO | ShaPO-T |
|---|---|---|---|---|
| Max HE ↓ | 25.2336 | 14.9509 | 9.4993 | **6.2466** |
| Win Rate ↑ | 42.82% | 69.19% | 82.50% | **85.45%** |

*Table 5.* Effect of reward supervision of ShaPO-R on LLaMA-3-8B. IID results are evaluated on PKU-SafeRLHF-30K, while OOD results report the average ASR across four OOD safety benchmarks.

| Methods | IID | | OOD (Avg.) | |
|---|---|---|---|---|
| | MD↓ | NV↓ | MD↓ | NV↓ |
| DPO | 26.57% | 14.41% | 28.78% | 16.91% |
| DPO+Reward | 18.42% | 8.40% | 17.33% | 7.02% |
| **ShaPO-R** | **16.32%** | **7.52%** | **15.98%** | **6.26%** |
| Dr.DPO | 4.39% | 0.50% | 3.72% | 0.45% |
| Dr.DPO+Reward | 3.38% | 0.38% | 2.25% | 0.35% |
| **ShaPO-R** | **1.13%** | **0.13%** | **1.30%** | **0.09%** |

flattening the loss geometry within this critical subspace.

**Effect of Reward Supervision.** Table 5 investigates whether ShaPO's robustness gains stem from reward supervision or optimization geometry. We instantiate ShaPO-R under both DPO and Dr.DPO alignment objectives and conduct a controlled ablation to isolate the effect of external reward signals. Two key observations emerge: (1) Under DPO, reward supervision yields substantial improvements, and ShaPO-R further reduces ASR across all settings, demonstrating complementary benefits. (2) Under Dr.DPO, reward gains become marginal, yet ShaPO-R still achieves consistent improvements, suggesting that geometry control provides orthogonal gains beyond what data-level signals can offer. This contrast reveals a fundamental distinction: reward-based and data-centric methods(such as Dr.DPO) operate at the data level, whereas ShaPO improves robustness at the parameter level via selective geometry control. These results confirm that ShaPO's gains are attributable to improved optimization geometry rather than stronger supervision, and that ShaPO serves as a general optimization framework that can be flexibly instantiated under diverse alignment objectives, including reward-based settings.

## 5. Conclusion

We revisit robustness in LLM safety alignment from the perspective of optimization geometry. We show that robustness failures in preference-based alignment cannot be fully explained by data uncertainty alone, and propose ShaPO, a geometry-aware preference optimization framework that selectively stabilizes alignment-critical parameter directions. By enforcing robustness where safety behavior is concentrated, ShaPO avoids the over-regularization of uniform

geometry control while improving robustness under domain shift and noisy supervision. Instantiated at both token and reward levels, ShaPO consistently strengthens safety robustness and composes with data-centric methods, highlighting optimization geometry as a complementary axis for robust LLM safety alignment. While ShaPO introduces additional computational overhead due to geometry-aware optimization, improving the efficiency and scalability of such approaches remains an important direction for future work, particularly for larger-scale models. In addition, although the identification of alignment-critical directions is necessarily based on approximate signals, our results suggest that even coarse geometric cues can meaningfully improve safety robustness.

## Acknowledgements

This research/project is supported by the National Research Foundation, Singapore under its National Large Language Models Funding Initiative (AISG Award No: AISG-NMLP-2024-002), the NExT++ Research Centre and ST Engineering under a collaborative research project (Agreement No. 2024-01813), and the National Natural Science Foundation of China (Grant No. U23B2031, 62436003). Any opinions, findings and conclusions or recommendations expressed in this material are those of the author(s) and do not reflect the views of National Research Foundation, Singapore.

## Impact Statement

This work focuses on improving the robustness of safety alignment in large language models. By making preference-based alignment less sensitive to small parameter perturbations, the proposed method aims to reduce unintended unsafe behaviors under distribution shift or noisy supervision. Like other optimization techniques, the impact of this work depends on how it is applied. While the method is designed for safety alignment, similar ideas could potentially be used for other objectives. We therefore encourage using this approach in safety-oriented settings and in combination with existing responsible AI practices.

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

# A. Algorithm and Optimization Details

---

**Algorithm 1** ShaPO: Sharpness-aware Preference Optimization

---

**Require:** Preference dataset $\mathcal{D}=\{(x_i, y_i^w, y_i^l)\}_{i=1}^N$, and probe training dataset $\mathcal{D}_p = \{(x_i, y_i)\}_{i=1}^M$; initial policy model $\pi_\theta$, reference $\pi_{\text{ref}}$, reward model $\mathcal{R}_\phi$ (only for reward-level ShaPO); hyper-parameters: learning rate $\eta$, batch size $B$, SAM frequency $\tau$, radius $\rho$, top fraction $r$.

**Ensure:** Aligned policy model $\pi_\theta^\star$.

 1: **Step 0: Safety-Critical Neuron Identification**
 2: Train probe $\mathbf{p}$ on $\mathcal{D}_p$ via Cross-Entropy Minimization;
 3: Compute similarity score $s_j = |\langle v_j, \mathbf{p}\rangle|$ for all dimensions of value vector for each transformer block;
 4: $S \leftarrow$ indices of Top fraction w.r.t. $s_j$.
 5: **for** $t = 1$ to $T$ **do**
 6:     Sample mini-batch $\mathcal{B} \subset \mathcal{D}$ of size $B$.
 7:     **Step 1: Gradient on Safety Subspace**
 8:     $g_\mathcal{S} \leftarrow \nabla_{\theta_\mathcal{S}} \mathcal{L}_{\text{align}}(\theta; \mathcal{B})$.
 9:     **Step 2: Adversarial Perturbation (SAM-style)**
10:     $\epsilon_\mathcal{S} \leftarrow \rho \cdot g_\mathcal{S}/(\|g_\mathcal{S}\|_2 + 10^{-12})$.
11:     $\theta' \leftarrow \theta + \epsilon_\mathcal{S}$ {only $\mathcal{S}$ perturbed}
12:     **Step 3: Robust Update**
13:     **if** $t \bmod \tau = 0$ **then**
14:         $\nabla_\theta \leftarrow \nabla_\theta \mathcal{L}_{\text{align}}(\theta'; \mathcal{B})$.
15:         $\theta \leftarrow \theta - \eta \nabla_\theta$.
16:     **else**
17:         $\nabla_\theta \leftarrow \nabla_\theta \mathcal{L}_{\text{align}}(\theta; \mathcal{B})$.
18:         $\theta \leftarrow \theta - \eta \nabla_\theta$.
19:     **end if**
20: **end for**
21: **return** $\theta^\star \leftarrow \theta$.

---

# B. Related Works

**Preference-based Safety Alignment.** Safety alignment aims to ensure that language models avoid generating harmful or unsafe outputs beyond general instruction-following ability (Ouyang et al., 2022; Ji et al., 2023a). A dominant paradigm is post-training with human or model preferences, where models are optimized to favor safe responses over unsafe ones. Early approaches such as Reinforcement Learning from Human Feedback (RLHF) (Bai et al., 2022; 2023) rely on reward modeling and policy optimization, which incur high computational costs and can be sensitive to noisy supervision (Gao et al., 2023). To simplify the pipeline, Direct Preference Optimization (DPO) (Rafailov et al., 2023) reframes safety alignment as supervised learning on preference pairs, achieving competitive instruction-following and safety behaviors without explicit reward models or reinforcement learning. Building on this paradigm, recent work explores alternative formulations including group-relative or multi-objective optimization (Shao et al., 2024; Guo et al., 2025; Zhao et al., 2025a), as well as methods that explicitly target or extract safety-relevant decision boundaries within aligned models (Ferrand et al., 2025; Lee et al., 2024). Despite rapid progress, preference-based safety alignment methods still exhibit limited generalization under distribution shifts or adversarial prompts (Wei et al., 2024), raising concerns about robustness in real-world deployments.

**Data-centric Robust Safety Alignment.** Recent studies show that post-training paradigms for safety alignment often achieve only shallow robustness and rely on shortcut learning, leading to brittle behavior under distribution shift or adversarial prompting (Wei et al., 2024; Raghavendra et al., 2024; Lee et al., 2024; Zhu et al., 2025b). To address these issues, a growing body of work improves robustness from the data and supervision perspective under noisy, biased, or unreliable preference signals. One line develops *robust preference objectives* that explicitly model supervision uncertainty, including margin-based or calibrated formulations (Gheshlaghi Azar et al., 2023; Mitchell, 2023), noise-aware optimization (Chowdhury et al., 2024), and distributionally robust extensions (e.g., Dr.DPO (Wu et al., 2024), WDPO, KLDPO (Xu et al., 2026), DoRA (Zhu et al., 2025a) and BalDRO (Shao et al., 2026)). Another line explores *data selection and filtering* strategies using reward margins (Deng et al., 2025; Huang et al., 2025), external guidance (Zhang et al., 2025), learned criteria (Gao et al., 2025)

and implicit margins (Liu et al., 2026). While effective at mitigating noise and bias in supervision, these approaches primarily address robustness at the data level and implicitly assume stable optimization dynamics in parameter space, leaving parameter-space robustness largely unexplored.

**Geometry-based Robust Optimization.** Beyond data uncertainty, the robustness and generalization of neural networks are also shaped by optimization geometry, such as sharpness and sensitivity to parameter perturbations (Foret et al., 2020; Bahri et al., 2021; Xie et al., 2026). Prior work in supervised learning has shown that controlling worst-case loss in a neighborhood of the parameters can improve robustness, often by favoring flatter regions of the loss landscape. These geometry-based approaches typically apply uniform control across the parameter space, implicitly assuming isotropic sensitivity. However, recent analyses indicate that parameter sensitivity is highly anisotropic, with loss variations concentrating along a small subset of directions (Ghorbani et al., 2019; Mi et al., 2022; Zhong et al., 2022; Yun, 2025; Zhu et al., 2024). For example, researchers have introduced Fisher-guided geometry in model merging (Roy et al., 2025). Despite this, geometry-based robustness has not been systematically explored in preference-based safety alignment. In this setting, optimization-induced sensitivity can amplify residual noise or distribution shift in preference supervision. Our work addresses this gap by introducing geometry-aware robustness into preference optimization, selectively stabilizing alignment-critical parameter directions rather than enforcing uniform flatness.

## C. Theoretical Insight: Detailed Proofs

### C.1. PAC-Bayes Generalization Bound of ShaPO

We present a concise proof of Theorem 1, establishing the PAC-Bayesian generalization bound for ShaPO.

**Setup.** Let $\mathcal{L}_D(\theta)$ and $\mathcal{L}_S(\theta)$ denote the population and empirical alignment losses, respectively. We decompose parameters as $\theta = (\theta_S, \theta_R)$, where $\theta_S$ corresponds to alignment-sensitive directions and $\theta_R$ to residual parameters. Assume that: (i) $\ell(f_\theta(x), y) \in [0, 1]$; (ii) $\mathcal{L}_S$ is $L_R$-Lipschitz with respect to $\theta_R$; (iii) the alignment loss is locally convex in $\theta_S$ within radius $\rho$.

**PAC-Bayes framework.** For any prior $P$ and posterior $Q$ on perturbations $\epsilon = (\epsilon_S, \epsilon_R)$, the standard PAC-Bayes inequality for bounded losses gives, with probability at least $1 - \delta$:

$$\mathbb{E}_{\epsilon \sim Q}[\mathcal{L}_D(\theta + \epsilon)] \leq \mathbb{E}_{\epsilon \sim Q}[\mathcal{L}_S(\theta + \epsilon)] +$$
$$\sqrt{\frac{\mathrm{KL}(Q\|P) + \ln(1/\delta)}{2n}}.$$

We consider $Q$ supported on the subspace ball $B_S(\rho) \times B_R(\rho_R)$. Applying the Lipschitz condition in $\theta_R$ yields:

$$\mathcal{L}_D(\theta) \leq \max_{\|\epsilon_S\| \leq \rho} \mathcal{L}_S(\theta + \epsilon_S) + L_R \rho_R$$
$$+ \sqrt{\frac{\mathrm{KL}(Q\|P) + \ln(1/\delta)}{2n}}.$$

**Bounding the KL term.** Under a local Laplace approximation $\mathcal{L}_S(\theta + \epsilon_S) \approx \mathcal{L}_S(\theta) + \frac{1}{2}\epsilon_S^\top H_S(\theta)\epsilon_S$, choosing Gaussian prior $P = \mathcal{N}(0, \sigma^{-2}I)$ and posterior $Q = \mathcal{N}(0, (H_S + \sigma^{-2}I)^{-1})$ gives

$$\mathrm{KL}(Q\|P) \approx \frac{1}{2}\log\det(I + \sigma^2 H_S) \leq$$
$$\frac{1}{2}d_S \log(1 + \sigma^2 \lambda_{\max}(H_S)).$$

Then, we have:

$$\mathcal{L}_D(\theta) - \mathcal{L}_S(\theta) = O\left(\sqrt{\frac{d_S \log(1 + \sigma^2 \lambda_{\max}(H_S))}{n}}\right) + \varepsilon_R,$$

where $\varepsilon_R = L_R \rho_R$ accounts for non-sensitive parameters. At a local optimum, the subspace sharpness:

$$\mathrm{sharpness}(\theta_S) = \max_{\|\epsilon_S\| \leq \rho}\left[\mathcal{L}_{\mathrm{align}}(\theta + \epsilon_S) - \mathcal{L}_{\mathrm{align}}(\theta)\right]$$
$$\approx \frac{1}{2}\rho^2 \lambda_{\max}(H_S). \tag{16}$$

Finally, we obtain the generalization bound of ShaPO :

$$\mathcal{L}_D(\theta) - \mathcal{L}_S(\theta) = O\Big(\sqrt{\tfrac{d_S \, \text{sharpness}(\theta_S)}{\rho^2 n}}\Big) + \varepsilon_R,$$

which formally explains how minimizing subspace sharpness improves the generalization ability of ShaPO.

## C.2. Bypass Boundary Expansion

We now justify the robustness result in Eq. (20) of the main text.

**Local risk approximation.** At a stationary point $\nabla\mathcal{L}_{\text{align}}(\theta) \approx 0$, a second-order Taylor expansion gives

$$\mathcal{L}_{\text{align}}(\theta + \Delta\theta) \approx \mathcal{L}_{\text{align}}(\theta) + \tfrac{1}{2}\Delta\theta^\top H(\theta)\Delta\theta$$
$$\leq \mathcal{L}_{\text{align}}(\theta) + \tfrac{1}{2}\lambda_{\max}(H)\|\Delta\theta\|^2.$$

**Bypass boundary.** Let $\tau$ be the safety threshold. To guarantee $\mathcal{L}_{\text{align}}(\theta + \Delta\theta) \leq \tau$ for all $\|\Delta\theta\| \leq \delta$, it suffices that

$$\mathcal{L}_{\text{align}}(\theta) + \tfrac{1}{2}\lambda_{\max}(H)\delta^2 \leq \tau,$$
$$\Rightarrow \quad \delta \leq \sqrt{\tfrac{2(\tau - \mathcal{L}_{\text{align}}(\theta))}{\lambda_{\max}(H)}}.$$

Since ShaPO reduces $\lambda_{\max}(H_S)$ by flattening the alignment loss in subspace $\theta_S$, it enlarges the feasible bypass radius $\delta$, providing stronger robustness to parameter perturbations.

# D. Experimental Setup

## D.1. Benchmarks

We evaluate safety alignment robustness across multiple benchmarks spanning diverse safety domains. These benchmarks differ in task formulation, harm taxonomy, and distributional characteristics, allowing us to assess both in-distribution safety performance and robustness under domain shift or adversarial prompts. Below, we summarize the covered safety domains, benchmark characteristics, and evaluation protocols.

- **PKU-SafeRLHF-30K** (Ji et al., 2023b) [8]: This benchmark consists of preference pairs annotated for safety, and is commonly used for training and evaluating preference-based safety alignment. We use its held-out split to measure in-distribution safety performance, reporting attack success rates under standard evaluation prompts.

- **HH-RLHF-Safety** (Bai et al., 2022) [9]: It's derived from the Anthropic HH-RLHF dataset, this benchmark focuses on the trade-off between helpfulness and harmlessness. It evaluates whether models appropriately refuse unsafe instructions while maintaining alignment with benign user intent.

- **Do-Not-Answer** (Wang et al., 2023b) [10]: This benchmark evaluates abstention behavior, measuring whether models correctly refuse to answer queries that should not be responded to. It emphasizes refusal precision under clearly unsafe requests.

- **HarmBench** (Mazeika et al., 2024) [11]: HarmBench consists of adversarially constructed prompts designed to induce harmful responses across multiple categories. It tests robustness to malicious intent and prompt-based attacks beyond the training distribution.

- **SaladBench** (Li et al., 2024) [12]: SaladBench focuses on compositional and obfuscated harms, where unsafe intent is embedded within multi-step or indirect queries. This benchmark evaluates whether models can maintain safety under more subtle or context-dependent threat scenarios.

---

[8]https://huggingface.co/datasets/PKU-Alignment/PKU-SafeRLHF-30K

[9]https://huggingface.co/datasets/asparius/antropic-hh-rlhf_sft

[10]https://huggingface.co/datasets/LibrAI/do-not-answer

[11]https://huggingface.co/datasets/walledai/HarmBench

[12]https://huggingface.co/datasets/walledai/SaladBench

## D.2. Baseline Description

We compare ShaPO against representative preference-based alignment methods from the DPO family, including DPO (Rafailov et al., 2023), cDPO (Mitchell, 2023), IPO (Gheshlaghi Azar et al., 2023), rDPO (Chowdhury et al., 2024), and Dr.DPO (Wu et al., 2024). These methods differ in how they model preference uncertainty and robustness, while sharing a common preference-optimization framework. Specifically, DPO serves as the standard baseline that directly optimizes the policy without an explicit reward model. cDPO and rDPO improve robustness to noisy preference labels via conservative or noise-aware objectives, respectively. IPO adopts an alternative preference-learning formulation without binary classification, while Dr.DPO incorporates distributional robustness into the DPO objective. All baselines are implemented using their original formulations and evaluated under the same training and evaluation settings for fair comparison.

## D.3. Implementation Details

For all preference-based optimization methods, we set the common hyperparameter $\beta = 0.1$ and the learning rate $lr = 5 \times 10^{-7}$. For Dr.DPO, an additional hyperparameter $\beta'$ is set to $1.0$ as recommended in the original paper. For our proposed *ShaPO*, we set the SAM optimization frequency $\tau = 5$ for both variants. For the reward-level ShaPO variant, we additionally set $\beta_r = 10$. For evaluation, WR is assessed using the GPT-4.1-mini API with temperature set to 0.7 to ensure consistent and reliable preference judgments. ASR is assessed using MD-Judge and NV-Judge, with decoding parameters set to temperature$= 0.0$ and top_p$= 1.0$ during inference.

| Method | Pythia-2.8B | LLaMA-3.2-3B | LLaMA-3-8B | Qwen-2.5-7B |
|---|---|---|---|---|
| DPO Series | 16 mins | 18 mins | 46 mins | 44 mins |
| *ShaPO-T* | 21 mins | 24 mins | 89 mins | 85 mins |
| *ShaPO-R* | 30 mins | 35 mins | 99 mins | 95 mins |

*Table 6.* Running time comparison between DPO-series baselines and our proposed ShaPO variants across different backbone models.

**Computation Resources.** All experiments are conducted on NVIDIA H100 GPUs with the following configurations: for Pythia-2.8B and LLaMA-3.2-3B, we use 2 GPUs with a batch size of 128, and for LLaMA-3-8B and Qwen2.5-7B, we use 2 GPUs with a batch size of 32. For all settings, each global batch is formed by 2 micro-batches (i.e., gradient accumulation steps = 2). All methods share the same batch size and training epochs to ensure fair comparison. Here, we report the running time comparisons between DPO and ShaPO in Table 6.

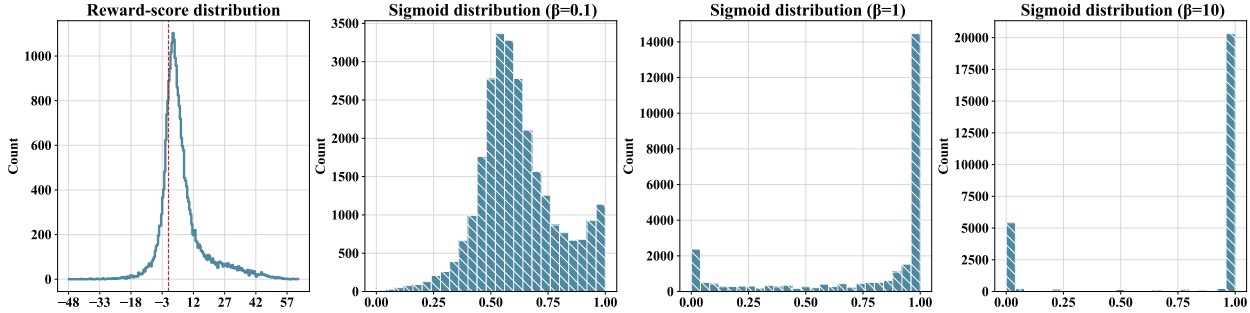

*Figure 6.* Reward-score distribution on the PKU-30K training set and the effect of different $\beta_r$ on score normalization. **Left**: the raw score difference $\Delta r = r(x, y^w) - r(x, y^l)$ produced by the Beaver reward (negated cost) model on preference pairs. **Right three**: the corresponding sigmoid-transformed values $\sigma(\beta_r \Delta r)$ under $\beta_r \in \{0.1, 1, 10\}$.

**Reward Model Usage.** In our reward-level ShaPO approach, we use a single safety-oriented preference model, `PKU-Alignment/beaver-7b-v1.0-cost`, trained on the PKU-SafeRLHF dataset. Although it is released as a *cost* model (where higher scores indicate more harmful outputs), we use it as a reward model by negating its outputs so that higher values correspond to more harmless responses. The resulting scalar reward signal is directly integrated into the ShaPO optimization objective, enabling explicit reward-level guidance during policy alignment. Therefore, we further analyze the reward signal on the PKU-30K training set: the distribution of the reward-score differences, as well as the resulting sigmoid-transformed distributions under different $\beta_r$, are shown in Figure 6.

# E. Additional Experimental Results

## E.1. Linear Probe Training

**Probe Accuracy.** We trained a linear safety classifier on the PKU-30k dataset, utilizing the "harmless" metric as the ground truth. In our implementation, the backbone language model remains frozen. We extract the residual stream representations from the final Transformer layer to serve as inputs for the linear classifier. Given a representation $h \in \mathbb{R}^d$ and a binary safety label $y \in \{0, 1\}$, the probe is optimized via a standard cross-entropy objective. We define the probe direction as $p = w_1 - w_0$, where $w_1$ and $w_0$ represent the class weights of the linear detector. Subsequently, we traverse the value vector to calculate the cosine similarity between each neuron and the probe vector. The top-$K$ neurons with the highest similarity scores are selected to form the safety parameter subspace.

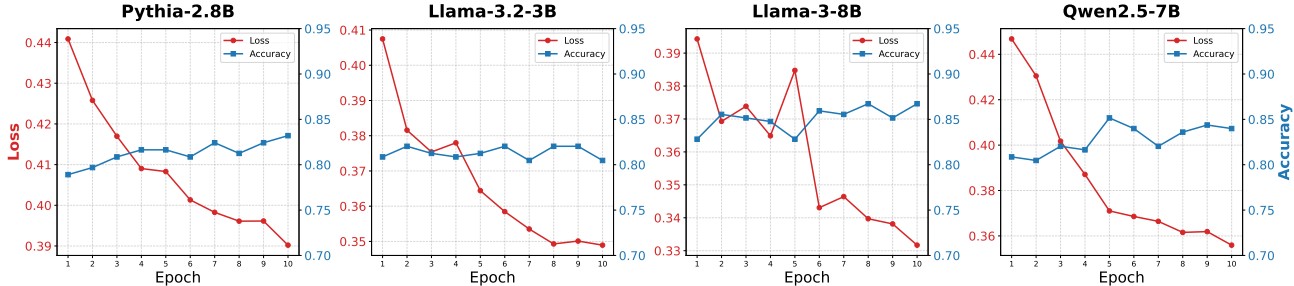

*Figure 7.* Probe training performance (Loss and Accuracy) across ten epochs for four different backbone models: Pythia-2.8B, LLaMA-3.2-3B, LLaMA-3-8B, and Qwen2.5-7B.

**Probe Decoding.** Table 7 presents the Top-20 decoded tokens associated with the trained probes across four different large language model backbones. For each backbone, we decode the tokens that are most strongly activated by the corresponding probe, which provides insight into the lexical patterns and semantic cues captured during probe training.

| Model | Probe to Tokens |
|---|---|
| Pythia-2.8B | unmist, **hypocr**, **bastard**, stad, Warning, hesis, warning, **illeg**, WO, ilis, %"}, heading, WARRANT, Processor, hurry, pretext, **culo**, ubation, untu, justification |
| LLaMA-3.2-3B | **Fuck**, **Fuck**, **fuck**, **hypocrisy**, **misogyn**, _stub, **fuck**, **dum**, dumpsters, **fucks**, **immoral**, **hypoc**, clums, **fucked**, **cuck**, **blowjob**, **FUCK**, Pressure, looph, weakest |
| LLaMA-3-8B | linger, Specifier, ORK, ÄL'n, .scalablytyped, **reesome**, LOPT, ijo, **fucks**, **stup**, èŃl', 404, akov, alim, ÐºÐ¾Ð³, imir, reich, VOID, stub, /common |
| Qwen-2.5-7B | juana, æīĠæľīæ̂ħèĬĤ, å̧ªªȩ̧̀ġæľī, æ¬£æħ°, orf, _DIGEST, .desktop, **culo**, âĬĶ, Stealth, xbe, unprotected, ÐºÑĠÐ¾Ð², roleName, ká°», .Apis, ä½ģ, blackColor, cont |

*Table 7.* Decoded top-20 tokens from the trained probes across four LLM backbones.

## E.2. Training Reward Margin

We report the training-time margin dynamics of both baseline methods and our approach across different LLM backbones, as shown in Figure 8. Overall, the proposed ShaPO family consistently exhibits the strongest margin growth while remaining compatible with the best-performing baselines, suggesting that selective geometry control effectively enhances preference separation.

## E.3. Safety Robustness under Domain Shift

In addition to the main results on larger backbones (Table 2), we evaluate safety alignment performance on smaller models, where limited capacity often leads to more brittle safety behaviors and higher attack success rates. Table 8 reports attack success rates (ASR) across four safety benchmarks, evaluated by two independent safety judges. Several consistent observations emerge. First, preference-based alignment methods substantially reduce ASR compared to vanilla

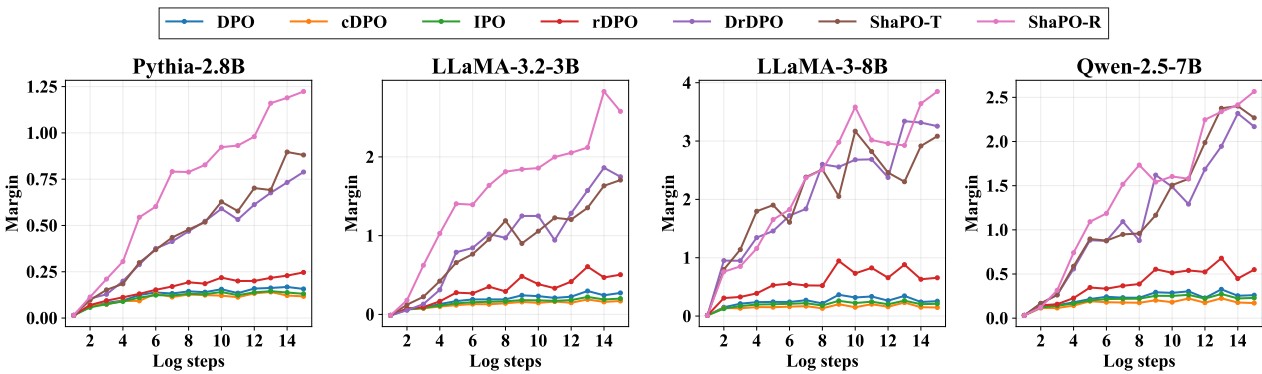

*Figure 8.* Comparison of training reward margins across different methods on four LLM backbones.

and SFT baselines, confirming the effectiveness of preference optimization for safety alignment. Second, data-centric robust objectives such as IPO, cDPO, rDPO, and Dr.DPO further improve safety performance on some benchmarks, but their gains are often inconsistent across judges and datasets. In contrast, ShaPO exhibits consistently low ASR across all benchmarks and both judges, indicating strong robustness to evaluation variance. Notably, ShaPO achieves large absolute reductions in ASR on smaller backbones such as Pythia-2.8B, where safety alignment is typically more challenging, while also maintaining near-zero ASR on stronger models such as LLaMA-3.2-3B. Finally, the close performance of ShaPO-T and ShaPO-R suggests that selective geometry control is robust to the specific choice of safety-critical subspace estimator, supporting the stability of the proposed approach. Overall, these results demonstrate that constraining parameter-space sensitivity leads to reliable safety improvements beyond data-centric robustness alone.

*Table 8.* Safety alignment performances on the domain shift setting, which were evaluated on four safety benchmarks with two judges. The reported results are conducted on Pythia-2.8B and LLaMA-3.2-3B backbones.

| Backbones | Methods | Do-Not-Answer | | HarmBench | | HH-RLHF | | Salad Bench | | AVG. |
|---|---|---|---|---|---|---|---|---|---|---|
| | | MD↓ | NV↓ | MD↓ | NV↓ | MD↓ | NV↓ | MD↓ | NV↓ | |
| Pythia-2.8B | Vallina | 59.70% | 18.34% | 86.00% | 29.50% | 72.02% | 24.95% | 78.95% | 29.69% | 52.30% |
| | SFT | 62.26% | 18.44% | 81.00% | 28.00% | 72.21% | 25.75% | 79.05% | 30.23% | 52.68% |
| | DPO | 46.26% | 14.50% | 65.00% | 15.00% | 58.82% | 21.06% | 63.60% | 20.79% | 41.21% |
| | IPO | 44.99% | 16.52% | 73.00% | 16.50% | 59.49% | 22.78% | 64.55% | 22.74% | 42.58% |
| | cDPO | 47.55% | 13.33% | 76.00% | 14.50% | 60.33% | 18.08% | 67.61% | 19.24% | 41.90% |
| | rDPO | 39.23% | 14.29% | 55.50% | 17.00% | 55.19% | 21.95% | 60.45% | 20.94% | 39.67% |
| | Dr.DPO | 16.42% | 9.81% | 25.50% | 10.00% | 29.35% | 19.94% | 30.27% | 18.24% | 23.90% |
| | *ShaPO-T* | 6.72% | **0.43%** | 10.50% | 1.50% | 14.14% | **1.67%** | 14.02% | **1.65%** | 7.71% |
| | *ShaPO-R* | **5.01%** | 0.64% | 11.50% | 0.50% | **12.39%** | 2.72% | **11.70%** | 2.00% | **6.91%** |
| LLaMA-3.2-3B | Vallina | 52.45% | 11.30% | 85.00% | 26.50% | 67.61% | 18.89% | 74.17% | 22.32% | 46.46% |
| | SFT | 51.71% | 27.72% | 89.50% | 73.00% | 66.93% | 38.67% | 74.28% | 50.58% | 59.27% |
| | DPO | 5.01% | 2.03% | 14.00% | 4.00% | 12.12% | 5.45% | 11.31% | 6.13% | 8.58% |
| | IPO | 3.73% | 1.71% | 17.00% | 6.00% | 12.10% | 5.26% | 11.12% | 6.07% | 8.45% |
| | cDPO | 11.51% | 5.54% | 65.50% | 15.00% | 24.55% | 10.67% | 26.65% | 13.29% | 19.12% |
| | rDPO | 1.17% | 0.64% | 3.50% | 1.00% | 5.92% | 2.02% | 4.19% | 1.97% | 3.27% |
| | Dr.DPO | 1.07% | **0.00%** | 2.00% | 0.50% | 2.57% | 0.34% | 1.78% | 0.33% | 1.20% |
| | *ShaPO-T* | **0.11%** | 0.11% | 1.50% | **0.00%** | **1.75%** | 0.23% | 1.21% | 0.25% | **0.59%** |
| | *ShaPO-R* | 0.75% | **0.00%** | **0.50%** | **0.00%** | 2.09% | 0.33% | **1.59%** | 0.3% | 0.99% |

