# OpenReview forum: "Revisiting Robustness for LLM Safety Alignment via Selective Geometry Control"
_ICML.cc/2026/Conference — ICML 2026 regular_

### Official Review · Reviewer_tSex · 2026-03-03

**Soundness:** 3
**Presentation:** 2
**Significance:** 2
**Originality:** 3
**Overall Recommendation:** 3
**Confidence:** 5

**Summary:**

This paper suggests a method to apply SAM to LLM for preference learning. Specifically, this paper selects safety-critical parameters for selective perturbation. For implementation, this paper suggests shapo-T(token) and shapo-R(reward). Empirically, it shows good robustness.

**Compliance With Llm Reviewing Policy:**

Affirmed.

**Final Justification:**

Thanks the authors for clarifying my concerns.

The authors’ clarification that Eq. 3 is merely "conceptual" confirms my concern about the fragmented and misleading presentation of the methodology.

I think the rebuttal still fails to demonstrate sufficient novelty, as applying a slightly modified SAM to sub-selected parameters remains an incremental contribution.

Therefore, I maintain my recommendation to reject.

**Key Questions For Authors:**

- What are the benefits of limiting the parameter perturbations to safety critical parameter subspace? Why should we do that?
- For limiting the number of parameters, there can be possibly differerent methods, e.g. only perturbing the last layer parameters. Why should we select the critical parameters as the similarity score as eq.(7)?
- Will this finding be only applicable to preference learning setting? or this can be applied to diverse situations where SAM is applied?
- Computations complexity?

**Limitations:**

See above.

**Strengths And Weaknesses:**

- What is the meaning of equation 3? For methodology, the method this paper suggests does not include data side robustness.
- I want ablation result for not using the soft label for shapo-r. Currently, it is not certain whether the performance gain is from soft label or from parameter perturbation.
- How the resulting loss surface really change (e.g. Hessian value?)?
- Why this study should be considered under the preference learning setting? I want to say that the good performances of SAM has been already well known. Why this SAM utilization should be considered as important under this setting?

---

> ### Author Rebuttal · Authors · 2026-03-31
>
> We thank the reviewer for the constructive feedback and insightful questions. Our key contribution is to show that robustness in safety alignment is **dominated by a small set of alignment-critical directions**, and that applying geometry control selectively to these directions is more effective than uniform SAM. We address the reviewer’s concerns point by point below.
>
> ---
>
> **W1: Meaning of equation(3).**
>
> Our work adopts a conceptual decomposition of robustness: prior approaches mainly focus on data-centric robustness (e.g., reweighting or denoising training samples), while ShaPO addresses parameter-side robustness by stabilizing optimization geometry. These two aspects are complementary. In fact, we show in Fig. 4 that ShaPO can be directly combined with data-centric methods (e.g., Dr.DPO), yielding further improvements. This demonstrates that our approach is not a replacement for data-side robustness, but a **composable and orthogonal enhancement**.
>
> ---
>
> **W2: Ablation of reward model.**
>
> We have addressed this question in our response to Reviewer ND4s. Please kindly refer to that section for details.
>
> ---
>
> **W3: How the resulting loss surface really change (e.g. Hessian value?)?**
>
> We thank the reviewer for the suggestion. We analyze the sharpness of the loss landscape by computing the maximum Hessian eigenvalue on the safety subspace:
>
> |Metric|Base|DPO|Dr.DPO|ShaPO-T (Full)|ShaPO-T (Selective)|
> |-|-:|-:|-:|-:|-:|
> |Max Hessian Eigenvalue|25.2336|14.9509|9.4993|7.5634|**6.2466**|
>
> As shown in this table, we have the following observations. In particular, ShaPO-T achieves the lowest maximum Hessian eigenvalue, indicating a significantly flatter loss landscape compared to existing alignment approaches. Notably, while full-parameter ShaPO-T improves overall flatness over Dr.DPO, it does not necessarily minimize curvature within the safety subspace. In contrast, the selective control (Top-1%) explicitly focuses optimization on this subspace and further reduces the curvature, suggesting that global flatness does not guarantee optimal geometry in safety-critical directions.
>
> ---
>
> **W4&Q3: Clarification of introducing SAM to preference learning.**
>
> The proposed idea is not specific to preference learning. The key insight is that robustness may be **dominated by task-relevant directions**, and thus geometry control should be applied selectively rather than uniformly. Preference learning is a natural setting where such structure is particularly evident due to noisy and ambiguous supervision. However, the same principle can extend to other tasks where identifying task-relevant directions is possible. We have also observed similar benefits of selective geometry control in other settings (e.g., deepfake detection), further suggesting that the idea is generally applicable beyond preference learning. **The main requirement is the ability to approximate such task-relevant directions.**
>
> ---
>
> **Q1: Benefits of selective geometry control vs full SAM**.
>
> Full SAM applies uniform geometry control, often over-regularizing irrelevant directions. In contrast, selective geometry control focuses on alignment-critical directions that drive robustness failures, thus improving robustness. Empirically, it outperforms full SAM, and performance degrades as the controlled subspace expands toward the full space (please kindly refer to the visualization of
>  [Sensitivity Analysis](https://anonymous.4open.science/r/shapo_sensitivity-4E6B) for details). This suggests that gains not only come from applying SAM but also from selective control.
>
> ---
>
> **Q2: Comparison with last-layer perturbation.**
>
> ShaPO does not rely on a specific subspace identification method, but requires capturing alignment-sensitive directions. We add comparison with a recent approach (ICLR 2026, *Sharpness-Aware Minimization in Logit Space Efficiently Enhances Direct Preference Optimization*), which applies SAM in the logit space (i.e., last layer). Its performance is consistently worse than ShaPO, indicating that **naive subspace identification is insufficient**.
>
> |Method|IID-MD|IID-NV|OOD-MD|OOD-NV|
> |-|-:|-:|-:|-:|
> |Dr.DPO|2.38|0.50|2.02|0.38|
> |ShaPO-T (Last-layer)|2.26|**0.13**|1.85|0.36|
> |ShaPO-T (Top-1% Subspace)|**1.25**|**0.13**|**1.32**|**0.24**|
>
> ---
>
> **Q3: Computations complexity.**
>
> ShaPO does not reduce per-step computation, as the forward–backward pass is still performed over all parameters, with masking applied during the update. In practice, we control the overhead by applying ShaPO-style updates **periodically (every τ steps)**, which significantly reduces the overall training cost while preserving its effectiveness (Appendix D.3). Empirically, we provide a comprehensive analysis of the ShaPO-style update frequency. Please refer to the response to Reviewer 7Nv2 for details.
>
> We appreciate further suggestions to help improve and strengthen this work！

---

> > ### Author Rebuttal · Reviewer_tSex · 2026-04-03
> >
> > My concerns are not fully solved as follow:
> >
> > 1. On Presentation (Eq. 3): The authors’ response misses the point. My concern was not whether they address both axes, but why Equation 3 is featured so prominently in the methodology when the proposed method does not consider the data side. This results in poor presentation and rather underlines the proposed method is rather fragmented. A methodology section should formally define what the paper does, not what it could have done. Including irrelevant terms in the core formulation is confusing and misleading for the reader.
> >
> > 2. On Contribution (SAM for Preference Learning): The authors’ claim that their method is applicable to diverse tasks (e.g., deepfake detection) undermines their contribution. It confirms that ShaPO is essentially a standard SAM applied to a sub-selected parameter space, rather than a novel optimization objective designed specifically for the unique challenges of preference learning. Simply applying a well-known tool (SAM) to a new domain as a modified version, although it is not adequately explained why it should be modified as such for this domian, does not meet the high bar for an ICML contribution.
> >
> > Additionally, directing a reviewer to "refer to a response for another reviewer" regarding a core methodological concern seems unprofessional and dismissive under this current openreview review system. Each reviewer’s independent concerns regarding the source of performance gains deserve a direct and self-contained response.

---

> > > ### Author Response · Authors · 2026-04-07
> > >
> > > Dear Reviewer tSex,
> > >
> > > Thank you for your detailed follow-up and for clarifying your concerns.
> > >
> > > Regarding Eq.(3), we would like to clarify that it is introduced as a conceptual decomposition of robustness to motivate the proposed framework, rather than as part of the method itself. We agree that its current presentation may lead to confusion about its role, and we will revise the manuscript to make this distinction clearer.
> > >
> > > Regarding the contribution, our goal is not to propose a new preference learning objective, but to introduce a geometry-aware optimization framework based on safety-critical subspaces and worst-case optimization. This positioning was clarified in the rebuttal, and we will further improve the presentation to better reflect this intent.
> > >
> > > During the rebuttal, we provided additional empirical analyses to examine the source of performance gains. These results consistently support that the improvements arise from selective geometry control rather than generic uniform regularization.
> > >
> > > Finally, we acknowledge that some of our previous responses may not have been fully self-contained. We will ensure that all clarifications are presented more clearly and independently. We appreciate your feedback and will incorporate these suggestions to further improve the clarity and presentation of the paper.
> > >
> > > Best regards,
> > > The Authors

---

### Official Review · Reviewer_7Nv2 · 2026-03-09

**Soundness:** 2
**Presentation:** 2
**Significance:** 3
**Originality:** 2
**Overall Recommendation:** 3
**Confidence:** 2

**Summary:**

This paper studies robustness in LLM safety alignment under domain shift and noisy preference supervision. The main idea is to improve preference optimization by applying sharpness-aware updates only within a probe-identified safety-relevant parameter subspace, instead of regularizing the full parameter space uniformly. The paper introduces two variants, ShaPO-T and ShaPO-R, and evaluates them on PKU-SafeRLHF and several OOD safety benchmarks across multiple model backbones. The reported results show consistent improvements over DPO-style baselines in IID, OOD, and noisy-label settings.

**Compliance With Llm Reviewing Policy:**

Affirmed.

**Final Justification:**

The authors addressed some of my main concerns.

**Key Questions For Authors:**

1. Can the authors compare ShaPO-T against full-parameter SAM / uniform sharpness control across the main IID, OOD, and noisy-label settings, rather than only in a limited ablation? This would help clarify whether selectivity is the main source of improvement.

2. Can the authors provide a more direct evaluation of utility / over-refusal on benign prompts or standard helpfulness benchmarks? This would strengthen the practical safety claim.

3. For ShaPO-R, can the authors clarify the fairness of using Beaver-7B-v1.0-cost trained on PKU-SafeRLHF, and ideally provide a cleaner control experiment?

4. How sensitive is the method to the choice of the selected subspace size, probe seed, perturbation radius, and update frequency?

**Limitations:**

The paper discusses computational overhead and the approximate nature of the identified safety subspace, but it should more explicitly discuss utility / over-refusal trade-offs, dependence on the external reward model in ShaPO-R, and sensitivity to the probe-defined subspace.

**Strengths And Weaknesses:**

**Strengths**

- The paper addresses an important and timely problem: robustness of safety alignment under realistic shift and supervision noise.
- The proposed idea is simple and intuitive. Restricting sharpness-aware perturbations to a safety-relevant subspace is a reasonable and well-motivated design.
- The paper covers multiple backbones, multiple benchmarks.
- The paper includes useful supporting analyses, including selective-vs-uniform control.

**Weaknesses**

- The main methodological novelty is somewhat moderate. At a high level, the method combines sharpness-aware optimization with probe-based subspace selection and standard preference optimization objectives. This is useful, but not a major conceptual departure.
- The paper does not yet fully isolate whether the gains come specifically from *selective* control, as opposed to more generic sharpness regularization. The ablation for this point is relatively limited.
- ShaPO-R is harder to compare fairly against the main baselines because it uses an external reward model trained on PKU-SafeRLHF, which gives it a different supervision signal from standard DPO-family methods.
- The theoretical discussion is more heuristic than rigorous, and I do not view it as a strong theory contribution.
- The evaluation focuses heavily on safety metrics, but does not sufficiently address utility / over-refusal trade-offs on benign prompts.

---

> ### Author Rebuttal · Authors · 2026-03-31
>
> We thank the reviewer for the detailed and constructive feedback. We would like to clarify that our contribution is not simply combining SAM with subspace selection, but showing that robustness failures in safety alignment are **highly concentrated in a small set of directions**, and that applying geometry control only to these directions is both necessary and more effective than uniform SAM. We address the concerns below with additional empirical evidence.
>
> ----
>
> **W1: Novelty Clarification.**
>
> We agree that ShaPO builds on existing components (e.g., SAM and preference optimization). However, our key contribution is not the combination itself, but the observation that **robustness failures in safety alignment are dominated by a small set of directions**, and that geometry control should be applied selectively rather than uniformly. This perspective is not captured by prior SAM-style methods, which assume uniform regularization across parameters. Our empirical results consistently support that selectivity is necessary, rather than a minor modification.
>
> ---
>
> **Q2: W2&Q1 Selective Control vs. Full-parameter SAM.**
>
> We provide more comprehensive comparisons between ShaPO(selective geometry control) and full-parameter SAM (uniform geometry control) across IID and OOD settings.  Please kindly refer to the visualization of  [Sensitivity Analysis](https://anonymous.4open.science/r/shapo_sensitivity-4E6B) for details. The results show a clear non-monotonic trend: performance improves from very small subspaces, peaks at a small fraction (~1%), and degrades as it approaches the full parameter space (equivalent to full SAM). These results demonstrate that the gains are not due to generic sharpness regularization, but from controlling a **small set of alignment-critical directions**. While we have not yet included noisy-label comparisons in this setting, the consistent trend across IID and OOD already demonstrates that selective control is the key factor over uniform SAM.
>
> ---
>
> **W3&Q3: Ablation of reward model.**
>
> We have addressed this question in our response to Reviewer ND4s. Please kindly refer to that section for details.
>
> ---
>
> **W4: Theoretical contribution justification.**
>
> We agree that our analysis is heuristic and does not aim to provide formal guarantees. Instead, it offers a mechanistic understanding of why robustness in safety alignment is dominated by a small set of directions, motivating selective geometry control over uniform SAM. We thus position our contribution as a practically grounded insight, supported by consistent empirical evidence, rather than a formal theory result.
>
> ---
>
> **W5 & Q2: Lack of evaluation on utility/over-refusal trade-offs.**
>
> We thank the reviewer for the suggestion and have added evaluations on **MT-Bench and OR-Bench** using the LLaMA-3-8B backbone:
>
> |Method|MT-Bench (GPT-4 Judge,↑)|Strict Refusal ↓|Overall Refusal ↓|
> |-|-:|-:|-:|
> |Base|7.331|0.303%|15.618%|
> |DPO|**7.578**|0.227%|12.282%|
> |Dr.DPO|7.450|**0.0758%**|9.780%|
> |**ShaPO-T**|7.350|**0.0758%**|**8.643%**|
>
> On MT-Bench, ShaPO-T achieves competitive performance (7.350 vs. 7.450 for Dr.DPO), indicating that our method does not introduce additional utility degradation beyond standard alignment methods. On OR-Bench, ShaPO-T achieves **the lowest strict/overall refusal rate** compared with baselines, verifying that its safety gains are not due to overly conservative behavior. Rather than over-refusing, ShaPO-T selectively enforces safety constraints, leading to more precise and calibrated refusal decisions.
>
> ---
>
> **Q4: Lacking Sensitivity Analysis .**
>
> Thanks for your constructive suggestions. For the sensitivity analysis of the Top-K ratio and perturbation radius, please kindly refer to the response to the Reviewer ND4s, as well as the visualization of
>  [Sensitivity Analysis](https://anonymous.4open.science/r/shapo_sensitivity-4E6B). Here is the additional analysis of the ShaPO-style update frequency on LlaMA-3.2-3B backbone:
>
> |frequency τ|Time(mins)|IID-MD|IID-NV|OOD-MD|OOD-NV|
> |-|-|-:|-:|-:|-:|
> |τ=0 (Dr.DPO)|**18.3**|2.38|0.50|2.02|0.38|
> |τ=1|48.2|**1.21**|**0.12**|1.29|0.25|
> |τ=2|30.8|1.23|0.14|**1.28**|**0.24**|
> |τ=5|24.3|1.25|0.13|1.32|**0.24**|
> |τ=10|20.2|1.5|0.25|1.41|0.28|
> |τ=20|19.7|1.96|0.41|1.82|0.34|
>
> We can find that the ShaPO-style update frequency balances time cost and alignment performance. In our experiments, we set τ=5 for the trade-off.
>
> We appreciate further suggestions to help improve and strengthen this work！

---

> > ### Author Rebuttal · Reviewer_7Nv2 · 2026-04-01
> >
> > Thank you for the additional rebuttal. The new reward-only baseline and the Top-K / radius sensitivity analysis are useful, but I still do not think they fully establish selective geometry control as the main source of improvement.
> >
> > At this point, the evidence still seems consistent with a weaker interpretation: SAM-style geometry regularization itself helps preference optimization, while the extra gain from the specific probe-defined selective subspace is relatively modest. In particular, uniform/full-parameter control already provides most of the improvement, and selective control improves it further but only by a limited margin. Moreover, ShaPO-R remains harder to interpret cleanly because it introduces an external reward model trained on the same alignment domain. Finally, the identified subspace is still a heuristic proxy based on correlation with final-layer value directions, rather than a stable or causally validated safety mechanism.

---

> > > ### Author Response · Authors · 2026-04-03
> > >
> > > Dear Reviewer 7Nv2,
> > >
> > > Thank you for your thoughtful follow-up and for acknowledging the additional analyses in our previous rebuttal. We appreciate your consideration and provide further evidence to clarify these points.
> > >
> > > **1. Role of Selective Geometry Control.**
> > >
> > > We conduct additional analysis with a stronger backbone (Dr.DPO), providing more evidence that uniform (full-parameter) SAM is insufficient and that selective geometry control is crucial.
> > >
> > > |Top-k|IID-MD|IID-NV|OOD-MD(avg)|OOD-NV(avg)|
> > > |---|---:|---:|---:|---:|
> > > |0(DPO)|26.57|14.41|28.78|16.91|
> > > |0.01%|28.38|14.42|28.10(-2.36%)|16.23(-4.02%)|
> > > |0.1%|28.52|15.62|27.33(-5.04%)|16.04(-5.14%)|
> > > |0.5%|29.63|15.67|27.10(-5.84%)|15.08(-10.82%)|
> > > |1%|27.32|13.91|**26.31(-8.58%)**|**14.12(-16.49%)**|
> > > |10%|27.09|13.61|26.84(-6.74%)|14.72(-12.95%)|
> > > |100%(Full-SAM)|**25.56**|**13.03**|26.83(-6.78%)|15.12(-10.59%)|
> > >
> > > |Top-k|IID-MD|IID-NV|OOD-MD(avg)|OOD-NV(avg)|
> > > |:--|--:|--:|--:|--:|
> > > |0(Dr.DPO)|4.39|0.50|3.72|0.45|
> > > |0.01%|3.38|0.38|3.25(-12.63%)|0.38(-15.56%)|
> > > |0.1%|3.89|0.52|3.11(-16.40%)|0.45(0%)|
> > > |0.5%|3.64|0.35|2.91(-21.77%)|**0.30(-33.33%)**|
> > > |1%|2.76|**0.13**|**2.38(-36.02%)**|0.35(-22.22%)|
> > > |10%|2.76|0.21|2.91(-21.77%)|0.57(+26.67%)|
> > > |100%(Full-SAM)|**2.63**|0.25|3.08(-17.20%)|0.63(+40.0%)|
> > >
> > > Across both backbones, selective control (Top-1%) consistently outperforms full-parameter SAM, indicating targeted geometry control is more effective than uniform smoothing. As shown in [Topk-Sensitivity](https://anonymous.4open.science/r/shapo_sensitivity-4E6B/drdpo_top_k_sensitivity.png), performance peaks in a small subspace and degrades toward full SAM, showing robustness gains concentrate in limited directions.
> > > Under DPO, selective control achieves larger reductions than full SAM (e.g., **8.58% vs. 6.78%** on OOD-MD and **16.49% vs. 10.59%** on OOD-NV). Under Dr.DPO, the gap increases (**36.02% vs. 17.20%** on OOD-MD). These results suggest that focusing on safety-critical directions yields more effective robustness improvements than uniform regularization.
> > >
> > > **2. Clarification of Reward Model.**
> > >
> > > Regarding the role of the reward model, we aim to clarify that **ShaPO does not rely on reward supervision**, but instead provides a general optimization framework that can be instantiated under different alignment objectives, including reward-based settings. We have conducted detailed reward ablations under both DPO and Dr.DPO settings, which we briefly summarize here for clarity.
> > >
> > > |Method|IID-MD|IID-NV|OOD-MD(avg)|OOD-NV(avg)|
> > > |:--|--:|--:|--:|--:|
> > > |DPO|26.57|14.41|28.78|16.91|
> > > |DPO+Reward|18.42|8.40|17.33|7.02|
> > > |**ShaPO-R**|**16.32**|**7.52**|**15.98**|**6.26**|
> > > |Dr.DPO|4.39|0.50|3.72|0.45|
> > > |Dr.DPO+Reward|3.38|0.38|2.25|0.35|
> > > |**ShaPO-R**|**1.13**|**0.13**|**1.30**|**0.09**|
> > >
> > > Under DPO, reward models bring substantial improvements, and ShaPO-R further improves upon them. Under Dr.DPO, reward gains are marginal, while ShaPO still yields consistent improvements (e.g., 2.25 → 1.30 on OOD-MD). This suggests reward and data-centric methods act at the data level, whereas ShaPO improves robustness via parameter-side geometry. Thus, gains are not due to stronger supervision but improved optimization geometry.
> > >
> > > **3. Effectiveness of the Identified Subspace.**
> > >
> > > Regarding the nature of the identified subspace, we agree that the probe-based approach provides a heuristic approximation and do not claim it captures a causal safety mechanism. Instead, our goal is to identify a structured subset of directions that are most relevant to worst-case alignment behavior.
> > >
> > > We support this view with the following evidence. First, as shown in Figure 1 of the manuscript, a very small fraction of neurons (e.g., ~1%) accounts for the majority of worst-case alignment loss, indicating a strong concentration effect. Second, our Top-K sensitivity analysis shows that performance peaks at a small subspace and degrades as it approaches the full parameter space, suggesting that robustness is governed by a limited set of directions.
> > >
> > > |Metric|Base|DPO|Dr.DPO|ShaPO-T(Full)|ShaPO-T(Sel)|
> > > |:--|--:|--:|--:|--:|--:|
> > > |Max Hessian Eigenvalue|25.2336|14.9509|9.4993|7.5634|6.2466|
> > > |Win Rate|42.82%|69.19%|82.50%|84.57%|85.45%|
> > >
> > > Third, as shown in the table above, our Hessian analysis reveals a consistent trend across methods: DPO, Dr.DPO, and ShaPO all reduce the maximum Hessian eigenvalue on the safety subspace, and lower curvature is associated with better alignment performance. Notably, selective control achieves both the lowest curvature and the strongest robustness. These observations suggest that safety alignment may be related to reducing parameter sensitivity along a small set of alignment-relevant directions, and that a coarse but structured approximation of this subspace can already be effective in practice.
> > >
> > > **We thank the reviewer for the patience and valuable feedback. We hope our additional analyses address your concerns and look forward to your more feedback.**
> > >
> > > Best regards,
> > > The Authors

---

### Official Review · Reviewer_ND4s · 2026-03-10

**Soundness:** 4
**Presentation:** 4
**Significance:** 3
**Originality:** 3
**Overall Recommendation:** 5
**Confidence:** 1

**Summary:**

The paper proposes ShaPO, a geometry-aware preference optimization method that improves the robustness of LLM safety alignment under domain shift and noisy preferences by applying sharpness-aware optimization to safety-critical parameter subspaces.

**Compliance With Llm Reviewing Policy:**

Affirmed.

**Final Justification:**

Thank you for addressing my concern. I will keep my original score.

**Key Questions For Authors:**

Could the authors include or discuss a reward-based baseline without ShaPO and report the quality of the reward model used? This would help clarify how much of the improvement comes from the reward signal versus the proposed ShaPO optimization.
The paper fixes the Top-1% neuron selection when defining the safety-critical subspace, and it would be helpful to study how sensitive the results are to different subspace sizes.

**Limitations:**

yes

**Strengths And Weaknesses:**

The proposed ShaPO framework is well motivated, and the optimization formulation is clearly presented. Empirical results across multiple safety benchmarks generally support the main claims. However, the analysis of the safety-critical subspace identification remains relatively limited. The paper is well written and the overall structure is clear. The main originality lies in introducing selective geometry control for preference optimization and applying sharpness-aware optimization to safety-critical parameter subspaces. This combination and its application to safety alignment provide a meaningful perspective. The paper also addresses an important problem in LLM safety alignment, namely improving robustness under domain shift and noisy preference supervision.

---

> ### Author Rebuttal · Authors · 2026-03-31
>
> We sincerely thank the reviewer for the positive assessment and for recognizing the contributions of our work across multiple dimensions. Our proposed ShaPO improves robustness by applying geometry control only to **alignment-critical directions**, rather than uniformly across all parameters. We hope the additional clarifications and experiments further improve the clarity and confidence in our conclusions.
>
> ---
>
> **Q1: Ablation of reward model.**
>
> We agree this is important and have added a reward-only baseline without ShaPO under different data-centric alignment objectives:
>
> |Method|IID-MD|IID-NV|OOD-MD(avg)|OOD-NV(avg)|
> |-|-:|-:|-:|-:|
> |DPO|26.57|14.41|28.78|16.91|
> |DPO+Reward|18.42|8.40|17.33|7.02|
> |**ShaPO-R**|**16.32**|**7.52**|**15.98**|**6.26**|
> |Dr.DPO|4.39|0.50|3.72|0.45|
> |Dr.DPO+Reward|3.38|0.38|2.25|0.35|
> |ShaPO-R|**1.13**|**0.13**|**1.30**|**0.09**|
>
> Based on the above Table, we have the following observations:
>
> - On DPO alignment objective, adding an external reward model brings significant gains, and ShaPO-R can further improve upon it, showing complementary benefits.
> - On Dr.DPO alignment objective, which already incorporates data-centric robustness, adding a reward model provides only marginal improvement, while **ShaPO-R still yields clear gains**.
>
> These results suggest that external reward models and data-centric methods (e.g., Dr.DPO) both improve robustness at the **data-level** (e.g., mitigating noise or reweighting samples), and their effects partially overlap. In contrast, ShaPO provides **parameter-side robustness** via selective geometry control, leading to additional and consistent improvements. This further confirms that the gains from ShaPO are not attributable to stronger supervision, but to improved optimization geometry.
>
> ---
>
> **Q2: Sensitivity of subspace sizes.**
>
> Thanks for your constructive suggestions. We perform all sensitivity analyses on **DPO + ShaPO**, as it provides the cleanest setting to isolate the effect of geometry control without confounding factors from reward models or data-centric modifications, with all results conducted on the LLaMA-3-8B backbone:
>
> |Top-k|IID-MD|IID-NV|OOD-MD (avg)|OOD-NV (avg)|
> |-|-:|-:|-:|-:|
> |0 (DPO)|26.57|14.41|28.78|16.91|
> |0.01%|28.38|14.42|28.10|16.23|
> |0.1%|28.52|15.62|27.33|16.04|
> |0.5%|29.63|15.67|27.10|15.08|
> |1%|27.32|13.91|**26.31**|**14.12**|
> |10%|27.09|13.61|26.84|14.72|
> |100% (Full-parameter SAM)|**25.56**|**13.03**|26.83|15.12|
>
> Please kindly refer to the visualization of  [Sensitivity Analysis](https://anonymous.4open.science/r/shapo_sensitivity-4E6B), we have the following observations:
> - Compared with DPO, full-parameter SAM already brings a significant improvement, indicating that smoothing the loss landscape via adversarial perturbation helps mitigate sharp minima introduced by preference optimization.
> - Compared with full-parameter SAM, ShaPO-T exhibits a non-monotonic trend as the Top-k ratio increases: performance first improves and then gradually degrades toward the full-parameter SAM level. When the perturbation region is too small (e.g., Top-0.01%), the worst-case estimation is insufficient, leading to limited gains (close to DPO). As the subspace expands, the model better captures safety-critical directions and achieves optimal performance (around Top-1%). However, when Top-k continues to increase, the perturbation becomes less targeted and approaches global smoothing, which dilutes the focus on safety-relevant subspaces and thus reduces the advantage over full-parameter SAM.
>
> We also conduct a sensitivity analysis of the perturbation radius:
>
> |radius ρ|IID-MD|IID-NV|OOD-MD (avg)|OOD-NV (avg)|
> |-|-:|-:|-:|-:|
> |0|26.57|14.41|28.78|16.91|
> |1e-7|26.90|14.19|27.96|16.34|
> |1e-6|27.32|13.91|**26.31**|**14.12**|
> |1e-5|**26.49**|14.15|26.97|14.85|
> |1e-4|27.18|14.53|27.12|15.01|
> |1e-3|27.33|14.58|27.20|14.98|
>
> We find the performance relatively insensitive to the perturbation radius. We attribute this to the fact that moderate perturbations stay within the same local basin and the preference optimization objective stabilizes the update, making ShaPO robust to the choice of radius. This also suggests that the effectiveness of ShaPO mainly comes from where to perturb (i.e., the selected subspace), rather than the exact perturbation magnitude.
>
> We appreciate further suggestions to help improve and strengthen this work！

---

> > ### Author Rebuttal · Reviewer_ND4s · 2026-04-02
> >
> > Thank you for addressing my concern. I will keep my original score.

---

> > > ### Author Response · Authors · 2026-04-03
> > >
> > > Dear Reviewer ND4s,
> > >
> > > Thank you very much for your kind follow-up and for confirming that our rebuttal has addressed your concerns. We are truly encouraged by your positive assessment of our work.
> > >
> > > We sincerely appreciate your recognition of our optimization-geometry perspective and selective geometry control, as well as your acknowledgment of the empirical results and clarity of the paper. Your thoughtful feedback has been invaluable in strengthening both the presentation and overall quality of our work.
> > >
> > > Best regards,
> > > The Authors

---

### Official Review · Reviewer_hMaU · 2026-03-13

**Soundness:** 3
**Presentation:** 3
**Significance:** 3
**Originality:** 3
**Overall Recommendation:** 4
**Confidence:** 3

**Summary:**

This paper studies robustness in LLM safety alignment from an optimization geometry perspective. While current robust preference optimization methods primarily address uncertainty/noise in human preference data, the authors argue that optimization induced sharpness along safety‑critical parameter directions is an independent and underexplored source of uncertainty.

To address this, the paper proposes Sharpness aware Preference Optimization (ShaPO), which enforces worst case robustness over a small safety critical parameter subspace, rather than uniformly regularizing the entire model. The paper introduces both token‑level and reward‑level instantiations of ShaPO, and demonstrates consistent robustness improvements under domain shift and noisy preference supervision across multiple models and benchmarks.

**Compliance With Llm Reviewing Policy:**

Affirmed.

**Final Justification:**

The authors addressed my concerns, and I am maintaining my score.

**Key Questions For Authors:**

1. How stable is the identified subspace across various datasets and probing mechanisms?
2. Have the authors explored alternative subspace identification methods? How sensitive would ShaPO be to this choice?
3. How sensitive are the results to the Top‑K percentage and perturbation radius? Is there a principled way to choose these values?
4. I am curious to hear authors thoughts on usefulness of the proposed methods to other alignment objectives like truthfulness, reasoning consistency etc. Is there anything that is limiting these methods uniquely to safety alignment?

**Limitations:**

Yes

**Strengths And Weaknesses:**

Strengths:

1. The distinction between data‑side robustness and parameter‑side robustness is explained well. The paper convincingly argues that robustness failures in safety alignment cannot be fully explained by noisy supervision alone.

2. Applying sharpness aware optimization only to a safety critical subspace is a novel improvement over prior SAM‑style approach. Empirical evidence supporting safety localization strengthens this design choice.

3. The paper demonstrates that ShaPO complements data robust preference optimization methods, which increases its practical relevance.

Weaknesses:

1. The reliance on linear probes provides a practical signal but lacks strong causal justification. While empirically effective, it is unclear how stable the identified subspace is across datasets, prompts, or architectures.

2. The analysis is local and curvature based, which is reasonable, but the paper does not provide generalization or robustness guarantees beyond local sharpness arguments.

3. The effect of top‑K selection threshold, probe training quality, perturbation radius is not extensively explored. Further detailed discussion on these could help with practical implementation.

4. While selective SAM is cheaper than full SAM, a clear discussion of training overhead and scalability would strengthen the practical impact.

---

> ### Author Rebuttal · Authors · 2026-03-31
>
> We sincerely thank the reviewer for the insightful feedback. We are encouraged that the reviewer recognized our optimization-geometry perspective on robustness, as well as the contribution of selective geometry control. We address the concerns below with additional empirical evidence.
>
> ---
>
> **W1&Q1: Probe&subspace stability**.
>
> We would like to clarify that the probe is used as a **practical proxy to rank alignment-sensitive directions**, rather than a causal identification of safety mechanisms. Empirically, ShaPO shows the stability:
>
> - Across datasets/prompts: ShaPO consistently improves OOD benchmarks despite the probe being trained on PKU-SafeRLHF, indicating robustness to safety-critical subspace estimation.
> - Across architectures: consistent gains on Pythia, LLaMA, and Qwen suggest the effect is not model-specific.
> - Robustness to subspace choice: our detailed **Top-K Selective ShaPO** shows stable performance across a range of sizes(0.5%~10%). Besides, compared with a naive subspace choice (e.g., last-layer), our proposed ShaPO performs better across various settings(see responses to the Reviewer tSex). This indicates that subspace-based structured selection is a simple yet effective solution for robust optimization.
>
> Our findings show that the failures in safety alignment are highly anisotropic: a small subset of directions dominates unsafe behavior. As a result, ShaPO only requires a **coarse but structured approximation** of these directions, rather than precise or globally stable subspace identification.
>
> ---
>
> **W2: The analysis is local and lacks global robustness guarantees.**
>
> We agree that our analysis is local and does not provide global robustness guarantees. Our goal is instead to characterize optimization-induced fragility in safety alignment. In particular, we show that such fragility is **highly anisotropic**, where a small subset of directions dominates worst-case behavior (as also evidenced in Fig. 1, where a few identified neurons account for most of the worst-case loss). This directly motivates selective geometry control, which stabilizes these directions rather than uniformly flattening the full parameter space.
>
> ---
>
> **W3&Q3: Sensitivity and principles to Top-K selection and perturbation radius**.
>
> Thanks for your constructive suggestions. We perform all sensitivity analyses on **DPO + ShaPO**, as it provides the cleanest setting to isolate the effect of geometry control without confounding factors from reward models or data-centric modifications, and conduct all experiments on LLaMA-3-8B. Please kindly refer to the response to the Reviewer ND4s, as well as the visualization of
>  [Sensitivity Analysis](https://anonymous.4open.science/r/shapo_sensitivity-4E6B).
>
> For the parameter selection principle, the Top-K ratio is guided by Figure 1 in the paper, where we select the approximated minimal subset that captures sufficient worst-case loss contribution, enabling targeted optimization. The perturbation radius is relatively insensitive, as moderate perturbations remain within the same local basin and the preference objective stabilizes the update.
>
> ---
>
> **W4: Training overhead and scalability of ShaPO.**
>
> We clarify that selective SAM is not designed to reduce per-step computation, as the forward–backward pass is still performed over all parameters, with masking applied only during the update. In practice, we control the overhead by applying ShaPO-style optimization **periodically (every τ=5 steps)**, which significantly reduces training cost while preserving its effect (as described in Appendix D.3). We have added the trade-off analysis between time cost and performance. We will clarify this trade-off in the revision.
>
> ---
>
> **Q2: Sensitivity to other subspace identification methods.**
>
> ShaPO does not rely on a specific subspace identification method, but requires capturing alignment-sensitive directions. Our comparison with the last-layer perturbation also verifies the superiority of the probe-based subspace identification approach.
>
> ---
>
> **Q4: Application Potential to other alignment task.**
>
> We agree this is a really interesting question. The proposed method is not specific to safety alignment, but builds on the general principle of SAM-based robust optimization. The key difference is that selective geometry control depends on identifying **alignment-sensitive subspaces**. In safety alignment, such subspaces can be approximated via simple probes (e.g., achieving ~85% accuracy) and are supported by strong concentration effects, where a small number of neurons dominate worst-case alignment loss. We thus believe the framework is broadly applicable: given a principled way to identify alignment-sensitive directions (e.g., for truthfulness or reasoning), selective geometry control can be naturally extended beyond safety.
>
> We appreciate further suggestions to help improve and strengthen this work！

---

> > ### Author Rebuttal · Reviewer_hMaU · 2026-04-03
> >
> > The authors addressed by questions and I don't have any follow ups!

---

> > > ### Author Response · Authors · 2026-04-03
> > >
> > > Dear Reviewer hMaU,
> > >
> > > Thank you for your kind follow-up and for confirming that our rebuttal has addressed your concerns. We are glad that our clarifications were helpful and that the key points are now clearer.
> > >
> > > We appreciate your recognition of our distinction between data-side and parameter-side robustness and the design of selective geometry control, as well as your acknowledgment of the empirical validation and clarity of the paper. Your feedback has been valuable in improving our work.
> > >
> > >
> > > Best regards,
> > > The Authors

---

### Decision · Program_Chairs · 2026-04-30

**Decision:**

Accept (regular)

**Comment:**

This paper studies robustness in LLM safety alignment from an optimization-geometry perspective. The authors propose ShaPO, which applies sharpness-aware updates selectively within a safety-relevant parameter subspace, rather than uniformly across the full parameter space. The method is evaluated across multiple backbones and safety benchmarks under IID, OOD, and noisy-preference settings.

The paper addresses an important and timely problem, and the empirical evaluation is one of its main strengths. Several reviewers found the motivation meaningful and noted that the paper offers an interesting perspective by distinguishing parameter-side robustness from more standard data-side robustness considerations. The method is also practically relevant, especially because it composes naturally with data-robust objectives such as Dr.DPO.

The main concerns raised during review were about the degree of methodological novelty, whether the gains come specifically from selective control rather than more generic SAM-style regularization, the heuristic nature of the probe-based subspace identification, and the presentation of Eq. 3. I agree that the novelty is moderate rather than major, and that the framing around Eq. 3 should be improved. In particular, Eq. 3 currently risks giving the impression that a broader robustness formulation is part of the formal method, whereas the authors clarified in rebuttal that it is intended only as conceptual motivation.

At the same time, I found the rebuttal helpful and materially strengthening. The added comparisons with full-parameter SAM, the sensitivity analyses, the utility/over-refusal evaluation, and the curvature analysis all improve the paper. In particular, the rebuttal provides additional empirical support for the claim that selective control can outperform uniform SAM-style regularization in this setting. That said, one reviewer remained unconvinced that the new evidence fully rules out a weaker interpretation, namely that SAM-style geometry regularization provides the primary gain while selective control adds a further but more limited improvement. I think this reservation is reasonable to note, but on balance I still find the empirical case sufficiently strong.

Overall, although not all concerns were fully resolved, I believe the paper makes a solid empirical contribution on an important problem and was substantially strengthened during rebuttal. I therefore recommend a weak accept.

For the camera-ready version, the paper should revise the presentation of Eq. 3 so that it is clearly framed as conceptual motivation rather than part of the formal method. It would also strengthen the final version to incorporate the additional rebuttal evidence more directly, especially the comparisons with full-parameter SAM and the added sensitivity and curvature analyses.